# SAINT: Sequence-Aware Integration for Spatial Transcriptomics Multi-View Clustering

**Zeyu Zhu**[1]    **Ke Liang**[1]*    **Lingyuan Meng**[1]    **Meng Liu**[1]    **Suyuan Liu**[1]

**Renxiang Guan**[1]    **Miaomiao Li**[2]    **Wanwei Liu**[1]    **Xinwang Liu**[1]*

[1]National University of Defense Technology, Changsha, China

[2]Changsha College, Changsha, China

## Abstract

Spatial transcriptomics (ST) technologies provide gene expression measurements with spatial resolution, enabling the dissection of tissue structure and function. A fundamental challenge in ST analysis is clustering spatial spots into coherent functional regions. While existing models effectively integrate expression and spatial signals, they largely overlook sequence-level biological priors encoded in the DNA sequences of expressed genes. To bridge this gap, we propose SAINT (Sequence-Aware Integration for Nucleotide-informed Transcriptomics), a unified framework that augments spatial representation learning with nucleotide-derived features. We construct sequence-augmented datasets across 14 tissue sections from three widely used ST benchmarks (DLPFC, HBC, and MBA), retrieving reference DNA sequences for each expressed gene and encoding them using a pretrained Nucleotide Transformer. For each spot, gene-level embeddings are aggregated via expression-weighted and attention-based pooling, then fused with spatial-expression representations through a late fusion module. Extensive experiments demonstrate that SAINT consistently improves clustering performance across multiple datasets. Experiments validate the superiority, effectiveness, sensitivity, and transferability of our framework, confirming the complementary value of incorporating sequence-level priors into spatial transcriptomics clustering.

## 1    Introduction

Spatial transcriptomics (ST) technologies measure gene expression while preserving the spatial layout of tissue sections, enabling the study of molecular organization in space [39, 46, 61, 54]. A key analysis task is clustering spatial spots into biologically meaningful regions, which reveals tissue structures and spatially patterned gene programs. This task can be naturally viewed as a multi-view clustering problem, where each spot is associated with both a gene expression profile and spatial coordinates [52, 48, 44, 11, 29]. Specifically, gene expression captures cell identity and state, while spatial proximity informs local organization and neighborhood relationships. Effectively integrating these heterogeneous views is key to producing biologically meaningful clusters.

Early attempts on spatial transcriptomics clustering primarily relied on unsupervised methods applied to gene expression profiles alone, ignoring the spatial structure inherent in the data [45, 43]. To incorporate spatial context, later approaches introduced regularization terms or handcrafted spatial distances that penalize discontinuities across neighboring spots. While effective to some extent, these methods often struggle to generalize to irregular tissue geometries or heterogeneous microenvironments. More recently, graph-based neural models have become the dominant paradigm due to their flexibility in capturing complex spatial relationships[24, 25, 17, 53, 30, 58]. For example,

---

*Corresponding authors.

39th Conference on Neural Information Processing Systems (NeurIPS 2025).

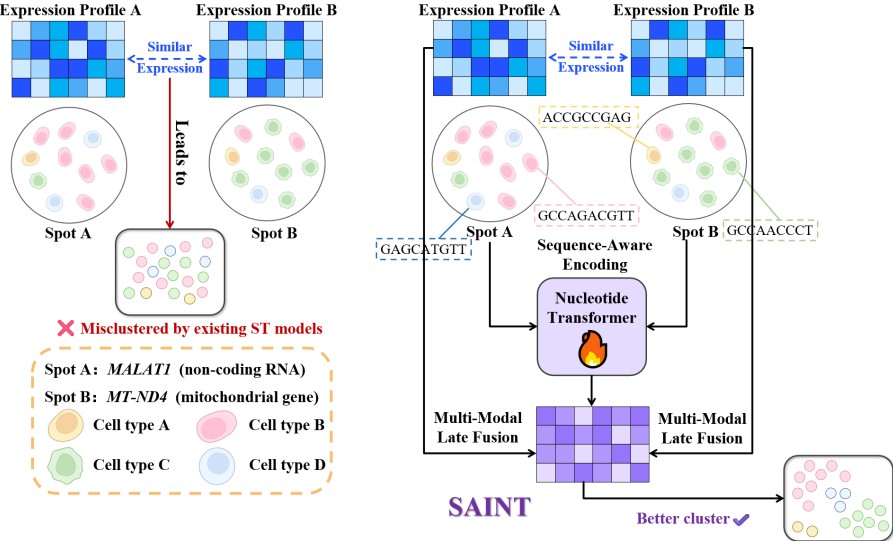

Figure 1: Motivation of SAINT. Existing ST clustering models may miscluster spatial spots with similar expression profiles but divergent gene functions. For example, Spot A and Spot B share highly similar expression profiles, yet express functionally divergent genes such as *MALAT1* and *MT-ND4*, respectively. This leads to incorrect clustering by expression-only models (left). By incorporating gene sequence features via a pretrained nucleotide encoder and multi-modal fusion, SAINT better distinguishes such cases and improves clustering accuracy (right).

SpaGCN [18] constructs a spatial graph based on physical coordinates and applies graph convolutions to jointly encode spatial proximity and gene expression. Building on this, Spatial-MGCN [47] aggregates multiple spatial graphs constructed from different biological priors (e.g., histological similarity, spatial distance), enabling a richer representation of local neighborhoods. Further, MAFN [62] proposes a late-fusion strategy that combines spatial, feature, and contrastive graphs through an adaptive attention module, improving clustering robustness under noise and sparsity. These GNN-based methods have consistently achieved state-of-the-art results across multiple ST benchmarks, highlighting the effectiveness of learning spot embeddings over spatial graphs.

However, existing ST clustering methods rely solely on observed gene expression and spatial proximity, while ignoring the rich biological information encoded in gene sequences. In practice, each spatial spot expresses a set of genes, and each gene is uniquely associated with a DNA sequence composed of nucleotides. These sequences often reflect regulatory elements or biochemical roles that are not evident from expression levels alone. As illustrated in Figure 1, spots A and B exhibit similar expression profiles. However, their dominant genes, *MALAT1* (a non-coding RNA) and *MT-ND4* (a mitochondrial protein-coding gene), indicate distinct biological roles. This discrepancy can lead clustering models to mistakenly group spatial spots into the same region simply because their expression profiles look similar, even though the expressed genes may have fundamentally different biological functions. Despite this, no prior work has systematically incorporated nucleotide-level sequence information into ST clustering, largely due to the following two key challenges in incorporating gene sequence knowledge into spatial transcriptomics clustering.

**(1) Lack of Sequence-Annotated Datasets.** Existing ST datasets do not contain nucleotide-level annotations, making it difficult to explore how gene sequences influence spatial gene expression or regional identity.

**(2) Cross-Modal Representation Integration.** Even with sequence information available, it remains unclear how to effectively encode gene sequences and integrate them with spatial and expression features for clustering. Naïvely combining modalities may lead to noise or semantic mismatch.

To fill this gap, we propose **SAINT** (Sequence-Aware Integration for Nucleotide-informed Transcriptomics), a sequence-informed framework that augments spatial transcriptomics with gene-level nucleotide embeddings. **First**, we construct sequence-augmented datasets across 14 tissue

sections spanning three widely used benchmarks: DLPFC (12 slices), HBC, and MBA. For each gene expressed within a spatial spot, we retrieve the corresponding reference DNA sequence from NCBI and organize the data into spotgenesequence mappings, enabling the integration of nucleotide-level information into the modeling pipeline. **Second**, we encode each DNA sequence using the Nucleotide Transformer, a large-scale pretrained language model for genomics. This yields rich, high-dimensional embeddings that capture regulatory signals and sequence-level semantics. These embeddings are projected into a lower-dimensional space and aggregated at the spot level to form sequence-derived spot representations. **After that**, we observe that each spot typically expresses dozens of genes, but not all are equally informative. To mitigate the influence of noisy or redundant sequences, we filter out low-variance genes and apply a lightweight attention mechanism to assign adaptive weights to the remaining gene embeddings, producing compact and spot-specific representations. **Finally**, the learned sequence-aware embeddings are integrated with expression and spatial features using a late fusion module. Experiments are conducted to evaluate the capacity of SAINT from four aspects: superiority, effectiveness, sensitivity, and transferability. The main contributions of our work are summarized as follows.

- **Problem.** Existing spatial transcriptomics (ST) clustering methods rely primarily on expression profiles and spatial coordinates, overlooking the biological priors encoded in gene sequences. This omission limits the semantic expressiveness of current representations and may lead to functionally mismatched clusters.

- **Dataset.** To address this, we construct multiple sequence-augmented ST datasets spanning 14 tissue sections from three benchmarks (DLPFC, HBC, MBA). Each spatial spot is annotated with the reference DNA sequences of its expressed genes, enabling the first systematic exploration of nucleotide-level information in ST clustering.

- **Method.** We propose SAINT, a sequence-informed multi-modal learning framework. SAINT encodes gene sequences using a pretrained genomic transformer model, filters uninformative genes based on expression variability, and applies attention-based aggregation to derive compact spot-level embeddings. These are then fused with expression and spatial representations through a lightweight late-fusion architecture.

- **Experiment.** Extensive experiments demonstrate that SAINT consistently improves clustering performance across multiple datasets. We evaluate SAINT from four perspectives: superiority, effectiveness, sensitivity, and transferability, confirming the complementary value of integrating nucleotide-level features into ST representation learning.

## 2 Related Work

This section summarizes recent related works from three aspects: (1) multiview clustering methods in spatial transcriptomics data, (2) genomic language modeling for sequence representation, and (3) sequenceaugmented spatial transcriptomics clustering. Due to space limitations, please refer to Appendix A.2 for a more detailed discussion.

## 3 Method

In this section, we present SAINT, a multi-modal learning framework for spatial transcriptomics clustering that integrates spatial coordinates, gene expression, and sequence-derived biological priors into unified spot-level embeddings. Unlike prior methods that rely solely on spatial and expression features, SAINT incorporates nucleotide-level representations to enhance spatial domain discovery. The framework of SAINT is shown in Fig.2.

The pipeline begins with data preprocessing, where each spatial spot is associated with its expression vector and the DNA sequences of its expressed genes. After normalization and selection of highly variable genes (HVGs), SAINT extracts complementary representations through two parallel modules. The structure-aware graph embedding module constructs spatial, feature, and combined graphs from HVGs, and encodes them using graph convolutional networks (GCNs). These multiview embeddings are then aggregated via attention to form a graph-derived representation $Z^{graph}$. In parallel, the sequence-aware encoder tokenizes DNA sequences and feeds them into a pretrained Nucleotide Transformer to produce gene-level embeddings, which are aggregated and projected into

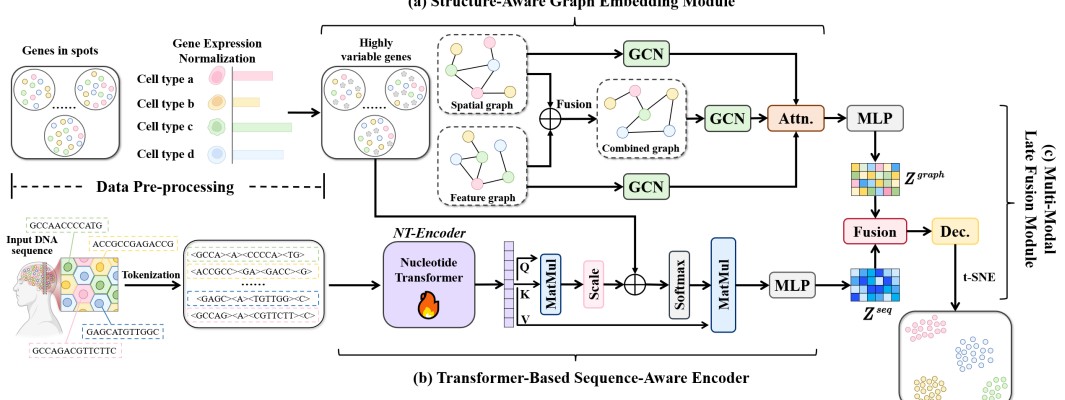

Figure 2: An overview of the SAINT framework. The pipeline starts with preprocessing, where each spatial spot is linked to its gene expression vector and DNA sequences. After normalization and HVG selection, two parallel modules extract structural and sequence-level features. (a) A graph embedding module encodes spatial, feature, and combined graphs via GCNs and fuses them through attention to obtain $Z^{\text{graph}}$. (b) A transformer-based encoder maps tokenized gene sequences to contextual embeddings, which are aggregated to form $Z^{\text{seq}}$. (c) The two representations are fused and passed to a ZINB decoder for expression reconstruction, and the resulting embeddings are used for clustering.

spot-level features $Z^{\text{seq}}$. The two representations are integrated via a late fusion module to obtain unified embeddings, which are then passed to a ZINB decoder for reconstructing gene expression. Clustering is performed in this fused latent space, capturing both structural topology and sequence semantics.

## 3.1 Problem Formulation

ST datasets consist of $N$ spatial spots, each profiled with transcriptome-wide gene expression and associated spatial coordinates. Formally, each spot $i \in 1, \ldots, N$ is described by three components: (1) a gene expression vector $\mathbf{x}i \in \mathbb{R}^G$ over $G$ genes, (2) a 2D spatial location $s_i \in \mathbb{R}^2$, and (3) a set of expressed genes $\mathcal{G}i = g_{i1}, g_{i2}, \ldots, g_{iM_i}$, where each gene $g_{ij}$ is associated with a DNA sequence $d_{ij}$ of variable length. These sequences are encoded into fixed-length embeddings using a pretrained genomic language model. The objective of spatial transcriptomics clustering is to partition the $N$ spots into $K$ spatial domains $\mathcal{C} = C_1, C_2, \ldots, C_K$, such that spots within the same cluster share similar expression programs, spatial context, and sequence-informed regulatory features. Unlike traditional transcriptomics clustering, this task involves integrating heterogeneous modalities, including gene expression, physical location, and sequence-derived embeddings, each with potentially different dimensionality, semantic structure, and noise characteristics.

Formally, our aim is to learn a unified embedding function $f : (\mathbf{x}_i, s_i, d_{ij}) \mapsto \mathbf{z}_i \in \mathbb{R}^d$ for each spot $i$, such that the resulting embeddings $\mathbf{z}_1, \ldots, \mathbf{z}_N$ can be effectively clustered into biologically meaningful domains. This requires (1) extracting semantically rich features from raw DNA sequences, (2) adapting to the varying number of genes per spot, and (3) designing a robust fusion strategy that preserves complementary signals while mitigating cross-modal redundancy.

## 3.2 Structure-Aware Graph Embedding Module

In parallel, SAINT models the spatial and expression views using graph-based encoders. We construct a spatial neighbor graph $\mathcal{G}_s = (\mathcal{V}, \mathcal{E}_s)$ where nodes correspond to spatial spots and edges connect neighboring spots based on 2D distance thresholding. We also build a feature graph $\mathcal{G}_f$ based on k-nearest neighbors in the gene expression space. Both graphs are encoded via GCNs.

To capture spatial proximity, expression similarity, and their interaction, we construct three graphs: the spatial graph $\mathcal{G}_s$, the feature graph $\mathcal{G}_f$, and the combined graph $\mathcal{G}_c$. All graphs share the same node set (spots), but differ in adjacency structure.

Each graph is encoded separately using a Graph Convolutional Network (GCN). Given a graph $\mathcal{G}_v$ with adjacency matrix $\mathbf{A}_v$ and node features $\mathbf{X}$, the GCN propagates features via as follows.

$$\mathbf{H}_v = \sigma\left(\tilde{\mathbf{D}}_v^{-1/2}\tilde{\mathbf{A}}_v\tilde{\mathbf{D}}_v^{-1/2}\mathbf{X}\mathbf{W}_v\right), \tag{1}$$

where $\tilde{\mathbf{A}}_v = \mathbf{A}_v + \mathbf{I}$ adds self-loops, $\tilde{\mathbf{D}}_v$ is the degree matrix, $\mathbf{W}_v$ is a trainable weight matrix, and $\sigma(\cdot)$ is a ReLU activation.

To model complementary topology between views, we define the combined graph $\mathcal{G}_c$ by aggregating the adjacency matrices of the spatial and feature graphs.

$$\mathbf{A}_c = \mathbf{A}_s + \mathbf{A}_f, \tag{2}$$

This combined structure captures both spatial continuity and expression similarity in a unified topology. The corresponding node embeddings $\mathbf{H}_s$, $\mathbf{H}_f$, and $\mathbf{H}c$ are then fed into an attention-based fusion module.

$$\mathbf{H}_{\mathrm{g}} = \mathcal{F}_{\mathrm{attn}}([\mathbf{H}_s \,\|\, \mathbf{H}_f \,\|\, \mathbf{H}_c]), \tag{3}$$

where $[\cdot, |, \cdot]$ denotes feature-wise concatenation, and $\mathcal{F}_{\mathrm{attn}}(\cdot)$ is an attention-based MLP that computes normalized weights across views to adaptively control the fusion process.

## 3.3 Transformer-Based Sequence-Aware Encoder

To capture biological priors at the sequence level, we employ a pretrained genomic language model, Nucleotide Transformer [10], to encode raw DNA sequences into dense vector embeddings. For each expressed gene $g_{ij}$ in spot $i$, we retrieve its reference DNA sequence $d_{ij}$ from curated databases (e.g., NCBI), and encode it into a dense vector.

$$\mathbf{z}_{ij} = \mathrm{NT\text{-}Encoder}(d_{ij}) \in \mathbb{R}^D, \tag{4}$$

where NT-Encoder$(\cdot)$ denotes the pretrained transformer model, and $D$ is the output dimension.

Since each spot may express a variable number of genes, we aggregate these gene-level embeddings into a fixed-dimensional representation using a lightweight attention pooling module. Specifically, for spot $i$, let $\mathbf{Z}_i = [\mathbf{z}_{i1}, \dots, \mathbf{z}_{iM_i}] \in \mathbb{R}^{D \times M_i}$ denote the matrix of gene embeddings. We compute attention weights $\alpha_{ij}$ over genes based on a softmax-normalized scoring function.

$$\alpha_{ij} = \frac{\exp(\mathbf{w}^\top \tanh(\mathbf{W}\mathbf{z}_{ij}))}{\sum_{j'=1}^{M_i} \exp(\mathbf{w}^\top \tanh(\mathbf{W}\mathbf{z}_{ij'}))}, \tag{5}$$

where $\mathbf{W} \in \mathbb{R}^{d_a \times D}$ and $\mathbf{w} \in \mathbb{R}^{d_a}$ are learnable parameters.

The sequence-derived embedding for spot $i$ is as follows.

$$\mathbf{z}_i^{\mathrm{seq}} = \sum_{j=1}^{M_i} \alpha_{ij} \cdot \mathbf{z}_{ij}, \tag{6}$$

To ensure compatibility with downstream fusion and to reduce computational overhead, we project the sequence embedding into a lower-dimensional space via a two-layer multilayer perceptron (MLP) with ReLU activation[16].

$$\mathbf{h}_i^{\mathrm{seq}} = \mathrm{MLP}_{\mathrm{seq}}(\mathbf{z}_i^{\mathrm{seq}}) \in \mathbb{R}^{d_s}, \tag{7}$$

where $d_s$ is a hyper-parameter denoting the projected dimension.

## 3.4 Multi-Modal Late Fusion Module

To obtain a comprehensive spot-level representation, we integrate information from both spatial graphs and gene sequences. Specifically, for each spot $i$, we concatenate the structural embedding $\mathbf{h}_i^{\mathrm{gcn}}$ with the sequence-derived embedding $\mathbf{h}_i^{\mathrm{seq}}$, capturing complementary features from spatial expression patterns and nucleotide-level signals. The concatenated vector is then projected into a shared latent space using a lightweight multilayer perceptron.

$$\mathbf{h}_i = \mathcal{F}_{\text{fuse}}([\mathbf{h}_i^{\text{gcn}}, |, \mathbf{h}_i^{\text{seq}}]), \tag{8}$$

where $[\cdot, |, \cdot]$ denotes concatenation, and $\mathcal{F}_{\text{fuse}}(\cdot)$ is a two-layer MLP with ReLU activation. This fusion module aligns the structural and sequence information into a unified representation, enabling the model to make better use of both expression and sequence-derived features during clustering.

## 3.5 Training Objective and Loss Functions

To jointly promote expression reconstruction, structural consistency, and cross-modal complementarity, we incorporate three loss components into the final training objective.

**ZINB Reconstruction Loss.** To model over-dispersion and dropout noise in ST data, we adopt a Zero-Inflated Negative Binomial (ZINB) decoder. The probability of observing count $x_{ig}$ for spot $i$ and gene $g$ is given as follows.

$$\text{ZINB}(x_{ig} \mid \mu_{ig}, \theta_{ig}, \pi_{ig}) = \begin{cases} \pi_{ig} + (1 - \pi_{ig}) \left( \frac{\theta_{ig}}{\theta_{ig} + \mu_{ig}} \right)^{\theta_{ig}}, & \text{if } x_{ig} = 0 \\ (1 - \pi_{ig}) \cdot \text{NB}(x_{ig} \mid \mu_{ig}, \theta_{ig}), & \text{if } x_{ig} > 0 \end{cases} \tag{9}$$

Here, $\mu_{ig}$ is the predicted mean expression, $\theta_{ig}$ is the dispersion parameter, and $\pi_{ig}$ models the dropout probability. $\text{NB}(\cdot)$ denotes the negative binomial distribution.

The total reconstruction loss over all $N$ spots and $G$ genes is computed as follows.

$$\mathcal{L}_{\text{ZINB}} = \sum_{i=1}^{N} \sum_{g=1}^{G} -\log \text{ZINB}(x_{ig} \mid \mu_{ig}, \theta_{ig}, \pi_{ig}), \tag{10}$$

where $x_{ig}$ denotes the observed count for gene $g$ in spot $i$.

**Graph Contrastive Regularization.** To preserve local structural smoothness in the learned embeddings, we encourage proximity between neighboring nodes and separation between non-neighbors via a contrastive loss.

$$\mathcal{L}_{\text{Reg}} = \mathbb{E}_{(i,j) \in \mathcal{N}} \left[ -\log \sigma(\cos(\mathbf{h}_i, \mathbf{h}_j)) \right] + \mathbb{E}_{(i,j) \notin \mathcal{N}} \left[ -\log(1 - \sigma(\cos(\mathbf{h}_i, \mathbf{h}_j))) \right]. \tag{11}$$

Here, $\mathcal{N}$ denotes neighbor pairs in the spatial or feature graph, $\cos(\cdot, \cdot)$ denotes cosine similarity, and $\sigma(\cdot)$ is the sigmoid function. $\mathbf{h}_i$ is the final embedding of spot $i$ after fusion.

**Cross-Modal Redundancy Reduction (DICR).** To encourage complementary information between the structural and sequence branches, we minimize feature redundancy using a decorrelation loss inspired by [33]. Let $C$ be the cross-correlation matrix computed between $\ell_2$-normalized embeddings from the two modalities. The loss is defined as follows.

$$\mathcal{L}_{\text{DICR}} = \sum_{i \neq j} C_{ij}^2 + \sum_i (C_{ii} - 1)^2, \tag{12}$$

where the first term penalizes off-diagonal correlations and the second term enforces identity alignment on the diagonal.

**Final Objective.** We jointly optimize the model by minimizing the following weighted sum of losses.

$$\mathcal{L} = \mathcal{L}_{\text{ZINB}} + \alpha \cdot \mathcal{L}_{\text{Reg}} + \gamma \cdot \mathcal{L}_{\text{DICR}}, \tag{13}$$

where $\alpha$ and $\gamma$ are hyper-parameters balancing the topology regularization and cross-modal decorrelation objectives. For consistency and fair comparison, we adopt the same hyperparameter configuration as MAFN [62], using its default values across all experiments without further tuning.

# 4 Experiment

We conduct comprehensive experiments to evaluate the performance and robustness of our SAINT across multiple dimensions, i.e., superiority, effectiveness, transferability, sensitivity and case Study. Specifically, we aim to answer the following five questions.

- **Q1: Superiority.** Does SAINT outperform existing state-of-the-art models on spatial transcriptomics clustering benchmarks?
- **Q2: Effectiveness.** How effective are the introduced sequence-aware augmentation strategies in enhancing clustering quality?
- **Q3: Transferability.** Can SAINT be flexibly integrated into different clustering backbones?
- **Q4: Sensitivity.** How sensitive is SAINT to variations in hyper-parameters?
- **Q5: Case Study.** Does SAINT produce biologically meaningful clustering results in real-world spatial transcriptomics datasets?

## 4.1 Experiment Setting

This section introduces the details of the experiment setting from four aspects, i.e., datasets, implementation details, compared baselines and evaluation metrics. Due to space limitations, details are provided in Appendix A.3.

Table 1: Clustering performance of competing spatial transcriptomics models. Bold entries indicate the best results, and underlined values denote the second-best.

| Method | Adjusted Rand Index (ARI) | | | | | | | | | |
|---|---|---|---|---|---|---|---|---|---|---|
| | 151507 | 151508 | 151509 | 151510 | 151669 | 151670 | 151671 | 151672 | HBC | MBA |
| SCANPY[49] | 0.20 | 0.15 | 0.19 | 0.14 | 0.10 | 0.09 | 0.12 | 0.12 | 0.49 | 0.23 |
| SpaGCN[18] | 0.43 | 0.33 | 0.41 | 0.37 | 0.23 | 0.35 | 0.51 | 0.53 | 0.56 | 0.34 |
| DeepST[51] | 0.55 | 0.42 | 0.43 | 0.50 | 0.44 | 0.33 | 0.52 | 0.48 | 0.53 | 0.25 |
| SCGDL[31] | 0.49 | 0.34 | 0.32 | 0.31 | 0.24 | 0.26 | 0.31 | 0.34 | 0.35 | 0.26 |
| stLearn[36] | 0.49 | 0.31 | 0.45 | 0.44 | 0.32 | 0.23 | 0.39 | 0.34 | 0.55 | 0.38 |
| Spatial-MGCN[47] | 0.63 | 0.46 | 0.54 | 0.51 | 0.39 | 0.35 | 0.60 | 0.77 | 0.64 | 0.42 |
| GraphST[34] | 0.48 | 0.49 | 0.52 | 0.50 | 0.48 | 0.46 | 0.61 | 0.63 | 0.54 | 0.41 |
| stMMR[56] | 0.59 | 0.51 | 0.58 | 0.69 | 0.49 | 0.48 | 0.68 | 0.63 | 0.62 | 0.44 |
| MAFN[62] | 0.68 | 0.51 | 0.71 | 0.61 | 0.56 | 0.48 | 0.82 | 0.76 | 0.60 | 0.43 |
| SAINT-G | 0.74 | 0.64 | 0.73 | 0.71 | 0.56 | 0.56 | 0.83 | 0.80 | 0.64 | 0.45 |
| SAINT-SA | **0.75** | **0.68** | **0.74** | **0.76** | **0.58** | **0.57** | **0.90** | **0.85** | **0.66** | **0.46** |
| Method | Normalized Mutual Information (NMI) | | | | | | | | | |
| | 151507 | 151508 | 151509 | 151510 | 151669 | 151670 | 151671 | 151672 | HBC | MBA |
| SCANPY[49] | 0.21 | 0.21 | 0.27 | 0.22 | 0.16 | 0.16 | 0.24 | 0.23 | 0.52 | 0.45 |
| SpaGCN[18] | 0.54 | 0.42 | 0.55 | 0.50 | 0.42 | 0.45 | 0.60 | 0.61 | 0.56 | 0.62 |
| DeepST[51] | 0.62 | 0.57 | 0.62 | 0.62 | 0.57 | 0.51 | 0.59 | 0.60 | 0.68 | 0.57 |
| SCGDL[31] | 0.55 | 0.44 | 0.48 | 0.45 | 0.38 | 0.36 | 0.41 | 0.46 | 0.43 | 0.64 |
| stLearn[36] | 0.64 | 0.53 | 0.62 | 0.59 | 0.49 | 0.41 | 0.54 | 0.47 | 0.63 | 0.66 |
| Spatial-MGCN[47] | 0.74 | 0.60 | 0.68 | 0.67 | 0.58 | 0.56 | 0.72 | 0.75 | 0.69 | 0.71 |
| GraphST[34] | 0.64 | 0.54 | 0.64 | 0.64 | 0.59 | **0.68** | 0.70 | 0.61 | 0.67 | 0.71 |
| stMMR[56] | 0.72 | 0.65 | 0.71 | 0.71 | 0.56 | 0.56 | 0.72 | 0.72 | 0.65 | 0.68 |
| MAFN[62] | 0.74 | 0.51 | 0.72 | 0.68 | 0.63 | 0.60 | 0.78 | 0.75 | 0.67 | 0.73 |
| SAINT-G | 0.77 | 0.69 | 0.73 | 0.72 | 0.62 | 0.63 | 0.78 | 0.79 | 0.69 | 0.72 |
| SAINT-SA | **0.78** | **0.71** | **0.74** | **0.73** | **0.64** | 0.64 | **0.84** | **0.80** | **0.70** | **0.74** |

## 4.2 Main Performance(RQ1)

To assess the effectiveness of SAINT, we compare it against nine state-of-the-art STC(Spatial Transcriptomics Clustering) methods on three benchmark datasets: DLPFC, HBC, and MBA. Table 1 reports ARI and NMI scores; full slice results are provided in Appendix A.4. SAINT includes two variants: **SAINT-G**, which averages gene-level nucleotide embeddings, and **SAINT-SA**, which incorporates sequence-aware attention for dynamic aggregation. As shown in Table 1, SAINT-SA consistently achieves the best or second-best results across nearly all slices, improving ARI from 0.64 to 0.68 and NMI from 0.69 to 0.71 on average. On slice 151507, SAINT-SA achieves an ARI

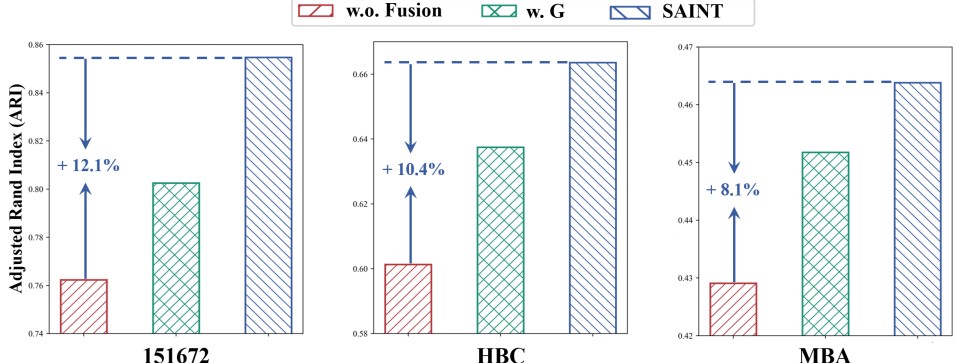

Figure 3: Ablation study of different components in SAINT.

Table 2: Clustering performance of competing spatial transcriptomics models. Bold entries indicate the best results, and underlined values denote the second-best.

| Method | 151507 | | 151508 | | 151509 | | 151510 | |
|---|---|---|---|---|---|---|---|---|
| | ARI | NMI | ARI | NMI | ARI | NMI | ARI | NMI |
| Spatial-MGCN | 0.6305 | 0.7443 | 0.4622 | 0.6023 | 0.5441 | 0.6812 | 0.5158 | 0.6668 |
| Spatial-MGCN+SAINT-G | 0.7125 | 0.7594 | 0.5888 | 0.6692 | 0.6766 | 0.6840 | 0.6189 | 0.6758 |
| Spatial-MGCN+SAINT-SA | **0.7357** | **0.7690** | **0.6476** | **0.6840** | **0.6986** | **0.7065** | **0.7060** | **0.7042** |
| MAFN | 0.6812 | 0.7402 | 0.5134 | 0.5183 | 0.7128 | 0.7213 | 0.6121 | 0.6822 |
| MAFN+SAINT-G | 0.7441 | 0.7723 | 0.6414 | 0.6933 | 0.7320 | 0.7279 | 0.7150 | 0.7165 |
| MAFN+SAINT-SA | **0.7471** | **0.7790** | **0.6843** | **0.7146** | **0.7426** | **0.7380** | **0.7649** | **0.7318** |

of 0.75 (+10.3% vs. MAFN), and an NMI of 0.84 on 151671 (+16.2% vs. GraphST). Notably, SAINT-G also surpasses most baselines. For example, it improves ARI by 8.8% over MAFN and by 10.3% on slice 151671. These results highlight the value of integrating nucleotide-level priors, even with simple aggregation. Additionally, SAINT maintains robust performance across both homogeneous (MBA) and heterogeneous (HBC) tissue contexts.

## 4.3 Ablation Study (RQ2)

To evaluate the contribution of different components in our framework, we conduct an ablation study comparing three model variants: (1) **w.o. Fusion**, a baseline without sequence integration; (2) **w. SA**, a variant that uses average-pooled gene sequence embeddings; and (3) **SAINT**, our complete model that incorporates sequence-aware attention and late-stage fusion. As shown in Figure 3, SAINT consistently outperforms the reduced variants across all benchmarks. For example, in terms of ARI, it yields relative improvements of +12.1% on slice 151672, +10.4% on HBC, and +8.1% on MBA. Even the average-pooling variant (*w. G*) surpasses the no-fusion baseline in most cases, indicating that sequence representations carry biologically meaningful information that complements expression-based features. Additional NMI results are reported in Appendix A.5.

## 4.4 Sensitivity Analysis (RQ3)

To evaluate the robustness of SAINT under different sequence embedding dimensions, we conduct a sensitivity analysis by varying $d_1$ and $d_2$, which represent the embedding dimensions used in SAINT-G and SAINT-SA, respectively. Each is selected from [16, 32, 64, 128, 256], and experiments are performed on three representative datasets, i.e., DLPFC (151507 as an example), HBC, and MBA. The ARI results are shown in Figure 4. Specifically, on the 151507 data slice, ARI fluctuates between 0.7351 and 0.7471, with a relative deviation of only 1.63%. For HBC, ARI varies from 0.6289 to 0.6433, corresponding to a 2.30% range. On the more challenging MBA dataset, the ARI spans 0.4269 to 0.4623, yielding a slightly wider but still manageable fluctuation of 3.30%. These small variations confirm that SAINT delivers stable clustering performance without requiring extensive hyperparameter tuning. Due to space constraints, the corresponding sensitivity results for NMI are reported in Appendix A.6, which exhibit similar trends.

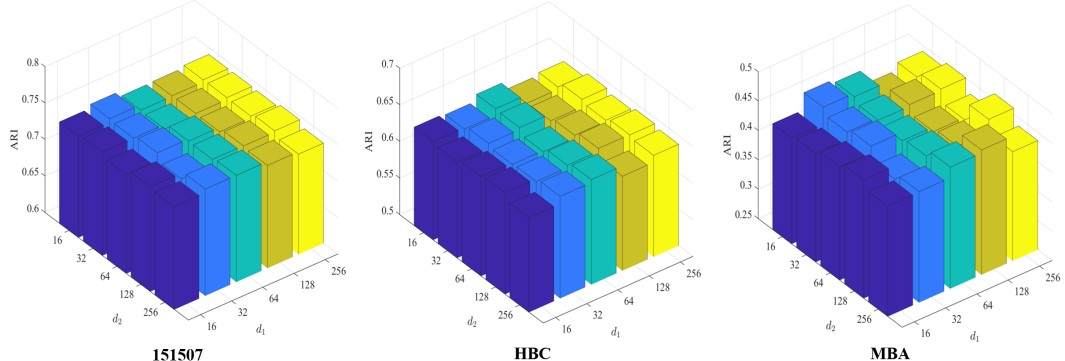

Figure 4: Parameter sensitivity analysis of the proposed SAINT on DLPFC, HBC and MBA datasets.

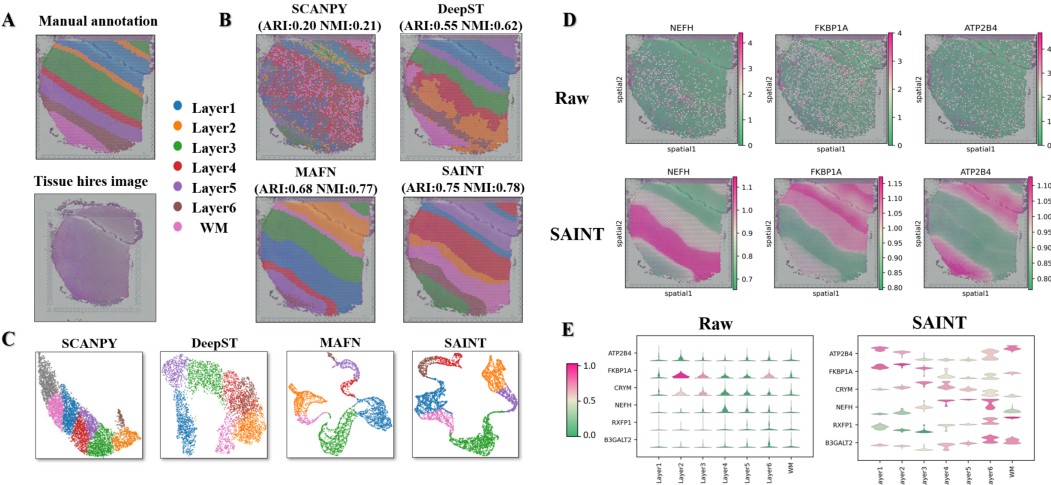

Figure 5: Case study visualizations on DLPFC slice 151507. (A) Manual tissue annotation overlaid on histological image. (B) Comparison of spatial domain identification results produced by competing methods. (C) UMAP projection of learned embeddings. (D) Spatial expression patterns of representative marker genes before and after SAINT enhancement. (E) Violin plots illustrating gene-level expression differences across identified domains.

## 4.5 Transferability Analysis (RQ4)

We evaluate the transferability of SAINT by integrating it into two representative backbonesSpatial-MGCN and MAFN. As shown in Table 2, both SAINT variants consistently improve clustering performance. For example, Spatial-MGCN+SAINT-G achieves an average ARI gain of +6.2%, while SAINT-SA further boosts it to +8.3%. On slice 151508, SAINT-SA raises the ARI to 0.6476, a relative improvement of +40.2% over the baseline (0.4622). Similar gains are observed with MAFN. On slice 151509, MAFN+SAINT-SA achieves an NMI of 0.7380, outperforming MAFN+SAINT-G by 1.67% and the vanilla MAFN by 2.93%. These results demonstrate that SAINT functions as a transferable and effective plug-in module. Meanwhile, this generality arises from the models modular design, where the sequence-aware encoder and cross-modal fusion can be seamlessly attached to existing spatial frameworks without retraining from scratch. Such adaptability highlights SAINTs potential as a unifying layer for future multi-modal spatial transcriptomics methods. Additional analysis is provided in Appendix A.7.

## 4.6 Case Study (RQ5)

We conduct a case study on the DLPFC 151507 slice to assess the interpretability and biological relevance of SAINT. As shown in Fig. 5(C), SAINT more accurately recovers cortical layer bound-

aries compared to competing methods. For instance, it captures the transition between Layer 5 and Layer 6 in the lower-right region that is oversmoothed by DeepST and fragmented in SCANPY. To further validate biological plausibility, Fig. 5(D) shows spatial expression maps of representative marker genes (*NEFH*, *FKBP1A*, *ATP2B4*). SAINT enhances spatial coherence and alignment with anatomical structures. Violin plots in Fig. 5(E) also demonstrate sharper inter-layer specificity and lower intra-layer variance. Notably, *ATP2B4* and *CRYM* show clearer separation across domains, while *B3GALT2* displays improved compactness within WM. These results highlight SAINTs ability to generate biologically meaningful and anatomically consistent representations. Additionally, this improvement mainly stems from the incorporation of nucleotide-informed embeddings, which enable SAINT to better distinguish functionally divergent genes with similar expression levels and thus refine boundary delineation. Such sequence-aware representations provide a mechanistic link between gene regulation patterns and observed spatial organization, further supporting the biological interpretability of the model. Additional case studies are provided in Appendix A.8.

## 5 Conclusion

In this work, we present SAINT, a sequence-aware multi-modal framework for spatial transcriptomics clustering. Unlike previous methods that utilize only gene expression and spatial proximity, SAINT introduces gene-level nucleotide embeddings to capture additional biological priors. To enable this integration, we construct spotgenesequence mappings across three benchmark datasets and encode sequences using a pretrained genomic language model. We design an attention-based aggregation module to summarize sequence features at the spot level, and employ a late fusion strategy to combine them with spatial-expression embeddings. Extensive experiments across multiple datasets demonstrate that SAINT consistently improves clustering accuracy.

## Acknowledgments

This work was supported by the National Key R & D Program of China No.2022YFA1005101, the National Natural Science Foundation of China (project No.61872371, 62032024, 62325604, 62441618, 62276271, 62506371).

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

# A  Appendix

## A.1  Limitations

While SAINT demonstrates strong performance and broad applicability, several limitations remain. First, the reliance on pretrained genomic models, such as the Nucleotide Transformer, may introduce biases from the training corpus, which predominantly includes well-annotated genes and reference sequences. This could limit generalizability to under-characterized genes or non-model organisms. Second, our current framework treats all expressed genes equally during sequence aggregation after HVG filtering. Although attention mechanisms mitigate some noise, rare but biologically significant genes may still be down-weighted or omitted. Future work may explore more context-aware gene selection strategies.

## A.2  Related Work

In this section, we summarize related work along three aspects. First, we review spatial transcriptomics clustering methods through the lens of multi-view learning, highlighting how expression and spatial features have been combined. Second, we introduce genomic language models that extract meaningful representations from DNA sequences. Finally, we discuss recent efforts toward integrating sequence information into spatial clustering, which remains an underexplored yet promising direction.

**MVC in Spatial Transcriptomics Data.** Clustering is a central task in spatial transcriptomics (ST), aiming to delineate spatially coherent tissue domains that reflect underlying biological structure and organization. Existing methods for ST clustering can be broadly grouped into three paradigms. (1) Expression-based approaches [5, 14] perform unsupervised clustering purely based on transcriptional profiles, typically using K-means or community detection on PCA-reduced features. (2) Spatially regularized models [57, 1] augment these strategies with spatial smoothing, either through distance-based penalties or spatial Laplacians to encourage neighboring spots to share cluster assignments. (3) In contrast, graph neural network (GNN)-based methods [13, 18, 55, 62] explicitly model spatial structure via graphs and propagate information using neural message passing. Among these, GNNs have emerged as particularly effective due to their flexibility in modeling complex tissue architectures and incorporating multimodal inputs. STAGATE [13] introduces a graph attention autoencoder that jointly learns from spatial adjacency and gene expression similarity, achieving state-of-the-art clustering performance across multiple platforms. SpaGCN [18], in a similar spirit, integrates histological information into the spatial graph, enabling more anatomically consistent clustering via graph convolutions. Meanwhile, SOTIP [55] formulates a spatial multi-task framework that captures micro-environmental structure and intercellular context through local neighborhood graphs. Moreover, MAFN [62] leverages a multi-view fusion strategy, adaptively combining representations from both spatial graphs and gene-gene similarity graphs to enhance robustness and tissue-domain separation. While these methods have shown strong empirical performance, they are all fundamentally built upon expression-derived features and spatial graphs. They often overlook additional layers of biological prior knowledge, i.e., such as the regulatory or structural information embedded in gene sequences, which may influence spatial gene expression patterns but remain underexplored in current models.

**Genomic Language Modeling for Sequence Representations.** While spatial transcriptomics clustering has traditionally focused on expression-level and spatial features, another promising source of biological prior lies in the gene sequences themselves. DNA sequences encode regulatory, structural, and evolutionary signals that can influence gene activity and co-expression. Recent years have seen a surge in the development of genomic language models (GLMs), inspired by advances in self-supervised learning from natural language processing. These models treat nucleotide sequences as a form of structured text and learn contextualized embeddings using masked language modeling or next-token prediction objectives. For example, DNABERT [19] adapts the BERT architecture [12] to k-mer tokenized DNA sequences, showing strong performance on transcription factor binding prediction and enhancer classification. Building on this foundation, Nucleotide Transformer [10] scales the model size and diversity of training data to cover multiple species and larger sequence contexts, achieving robust generalization across downstream genomics tasks. Unlike traditional handcrafted motif features or one-hot encodings, GLMs capture long-range dependencies, compositional signals, and shared patterns across different genomic loci. These properties make them attractive for repre-

sentation learning in biological applications, especially when labeled data are limited. Moreover, the pretrained embeddings can be transferred and fine-tuned to serve various predictive tasks, such as variant effect prediction [4, 60, 38, 21], epigenomic state modeling [41, 59, 20, 42], and multi-omics integration [15, 27, 7, 6]. Despite these advances, the use of sequence-based embeddings in spatial transcriptomics remains largely unexplored. Existing ST models rarely utilize the nucleotide sequences of expressed genes, thereby missing the opportunity to incorporate regulatory priors that may underlie the observed expression patterns. A few recent efforts have begun exploring protein-level embeddings for single-cell data [32], but nucleotide-level integration in spatial contexts has not been systematically studied.

**Sequence-Augmented Spatial Transcriptomics Clustering.** Recent advances in genomic language modeling have enabled the extraction of rich sequence-level representations from raw DNA. While these sequence-level embeddings have shown utility in a range of genomics tasks, their incorporation into ST clustering remains largely unexplored. This subsection reviews recent advances in augmenting ST models with sequence-derived features. A straightforward approach is late fusion, where sequence features are concatenated with spatial representations prior to downstream prediction. This method is modular and simple to implement, and has been widely applied in multi-modal omics integration [3, 28]. However, naive concatenation may fail to capture interactions between modalities and cannot dynamically adjust to varying gene importance across spatial contexts. To mitigate this, attention-based mechanisms have been introduced to assign adaptive weights to gene embeddings based on their contextual relevance. In spatial omics, attention modules have been used to integrate expression profiles with histological context or neighborhood structure [18, 8]. Soft attention mechanisms have also been employed to aggregate gene-level embeddings within each spot, yielding compact and informative representations that emphasize contextually relevant sequences. Transformer-based architectures offer a more flexible alternative by modeling interactions across gene sequences through self-attention. Though effective in natural language and genomics [9, 19], such models demand substantial training resources and are not yet widely used in spatial transcriptomics. Lightweight alternatives include expression-weighted averaging, which gives more influence to highly expressed genes in embedding aggregation [40]. Filtering by HVGs provides another practical benefit, reducing redundancy and focusing on the most informative sequence signals [2]. Despite these developments, existing ST clustering methods rarely incorporate sequence-level priors, leaving untapped the potential of regulatory DNA features in spatial organization. To bridge this gap, we introduce a novel framework that integrates gene sequence embeddings into ST clustering.

### A.3 Experiment Setting

Experiment settings are introduced from four aspects, i.e., datasets, implementation details, compared baselines and evaluation metrics.

**Datasets**. We conduct experiments on three benchmark datasets commonly used in spatial transcriptomics clustering:

We evaluate our method on three publicly available and widely used spatial transcriptomics datasets, spanning both human and mouse tissues. These datasets provide diverse anatomical and pathological contexts for robust model evaluation.

**LIBD Human Dorsolateral Prefrontal Cortex (DLPFC) Dataset.** The DLPFC dataset, curated by the LIBD research group [35], provides high-resolution spatial transcriptomic profiles from post-mortem human brain tissue, generated using the 10x Genomics Visium platform. It comprises 12 sagittal tissue sections, each covering approximately 3,6004,000 barcoded spots with over 33,000 protein-coding genes profiled per section. These slices encompass the full laminar structure of the neocortex (L1L6) as well as the white matter beneath. The dataset includes expert-annotated spatial domains based on histological examination, facilitating benchmarking of computational clustering models.

**10x Visium Human Breast Cancer (HBC) Dataset.** The HBC dataset [50] contains spatially resolved gene expression measurements from human breast tumor sections. Each spot captures the expression of approximately 36,000 genes across diverse histological structures. The dataset is annotated with 20 spatial domains, encompassing pre-invasive ductal carcinoma in situ (DCIS), lobular carcinoma in situ (LCIS), invasive ductal carcinoma (IDC), tumor stroma, immune infiltration zones,

Table 3: Statistics of the constructed sequence-augmented datasets. Each dataset corresponds to a tissue section or benchmark. "#B. class" denotes the number of unique barcodes (spots), "#G. class" indicates the number of distinct genes with available nucleotide sequences, and "Num" refers to the total number of (`barcode, gene, sequence`) triplets after filtering.

| Dataset | #B. class | #G. class | Num | Dataset | #B. class | #G. class | Num |
|---------|-----------|-----------|--------|---------|-----------|-----------|--------|
| 151507 | 721 | 237 | 24,968 | 151672 | 578 | 319 | 15,551 |
| 151508 | 849 | 190 | 11,940 | 151673 | 465 | 423 | 16,855 |
| 151509 | 697 | 249 | 12,489 | 151674 | 373 | 561 | 20,542 |
| 151510 | 737 | 233 | 12,092 | 151675 | 572 | 386 | 14,486 |
| 151669 | 563 | 341 | 16,034 | 151676 | 517 | 382 | 14,606 |
| 151670 | 596 | 316 | 15,089 | HBC | 182 | 250 | 32,734 |
| 151671 | 531 | 330 | 15,999 | MBA | 182 | 300 | 40,729 |

and adjacent normal tissue. This rich spatial annotation supports fine-grained exploration of tumor heterogeneity and microenvironmental interactions.

**Mouse Brain Anterior Tissue (MBA) Dataset.** The MBA dataset [26] includes gene expression profiles from a sagittal-anterior section of the adult mouse brain, obtained using the Illumina NovaSeq 6000 platform. It contains 2,695 spatial spots and over 32,000 genes, capturing molecular patterns across anatomical regions such as the cortex, hippocampus, and basal forebrain. This dataset enables detailed investigation of spatial gene regulation and inter-regional signaling in a mammalian neural context. It is publicly available through the 10x Genomics data repository.

**Sequence-Augmented Triplet Construction.**

To integrate nucleotide-level information into spatial transcriptomics clustering, we construct a structured sequence-augmented dataset for each benchmark. For every expressed gene within a spatial spot (barcode), we retrieve its corresponding reference DNA sequence from the NCBI nucleotide database using standardized gene identifiers. Each data entry is formatted as a (`barcode, gene, sequence`) triplet. To ensure biological relevance and avoid noisy or sparse entries, we remove barcodes expressing fewer than 10 genes with valid sequences. This filtering step ensures that each spatial spot contributes sufficient nucleotide-level context for representation learning. Table 3 summarizes the resulting datasets. For each tissue slice or benchmark, we report the number of unique barcodes (#B. class), the number of distinct genes with matched sequences (#G. class), and the total number of triplets (Num). These triplets serve as the input for sequence encoder modules and enable the downstream modeling of spatial domains with nucleotide-informed priors.

**Implementation Settings.** All models are implemented in PyTorch 2.0.1 and trained using the Adam optimizer [22] on a workstation with an Intel Core i9-9900K CPU, 64GB RAM, and an NVIDIA RTX 3090 Ti GPU. Following MAFN [62], we adopt consistent training settings and learning rate schedules. For sequence embedding, we evaluate two aggregation variants:

- **SAINT-G**: gene sequence embeddings are averaged without expression weighting.
- **SAINT-SA**: expression-aware attention pooling is applied to gene embeddings per spot.

The projection dimensions $d_1$ (for SAINT-G) and $d_2$ (for SAINT-SA) are selected from $\{16, 32, 64, 128, 256\}$.

**Compared Baselines.** To comprehensively evaluate the effectiveness of SAINT, we compare it against a wide range of state-of-the-art spatial transcriptomics clustering methods, including both traditional approaches and recent GNN-based frameworks.

- **SCANPY** [49] is a widely used single-cell analysis toolkit that performs PCA-based dimensionality reduction and graphbased clustering on highly variable genes without incorporating spatial context.
- **SpaGCN** [18] constructs a spatial graph from tissue coordinates and employs graph convolution to jointly model gene expression and spatial dependencies for anatomically coherent clustering.

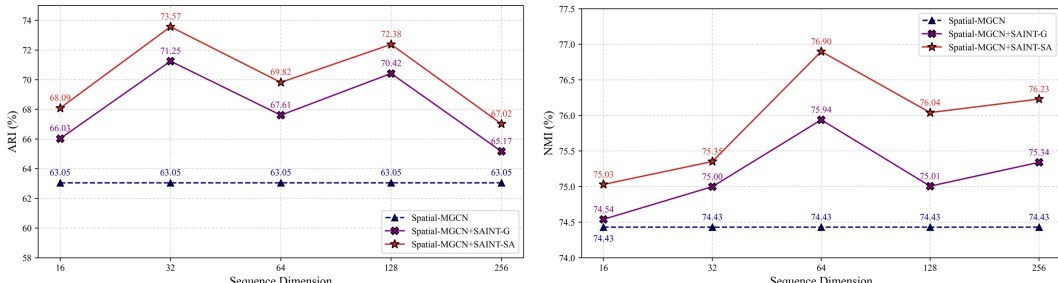

Figure 6: Transfer study of SAINT variants on the DLPFC 151507 slice under the Spatial-MGCN backbone.

- **DeepST** [51] learns spatially informed spot embeddings by jointly modeling transcriptional profiles and physical locations, enabling accurate tissue domain segmentation.

- **SCGDL** [31] utilizes a residual gated graph neural network augmented with a deep graph information maximization module to capture hierarchical and long-range dependencies in spatial transcriptomic graphs.

- **stLearn** [36] integrates histological morphology with gene expression and spatial coordinates through a self-supervised learning pipeline that regularizes clustering via spatially constrained random fields.

- **stMMR** [56] applies Markov random field regularization to smooth clustering labels and enforce spatial coherence in noisy or irregular tissue regions.

- **GraphST** [34] proposes a spatially guided contrastive learning framework that enhances intra-domain cohesion and inter-domain separation across spot embeddings.

- **Spatial-MGCN** [47] fuses multiple spatial and expression graphs using multi-view graph convolutions and an attention mechanism to capture heterogeneous tissue patterns.

- **MAFN** [62] employs a multi-branch graph convolutional architecture with adaptive late-fusion, allowing flexible integration of spatial and gene-gene similarity representations for robust clustering.

**Evaluation Metrics.** We adopt two widely used metrics for evaluating clustering performance:

**Adjusted Rand Index (ARI)** [37]: Measures similarity between predicted and ground-truth cluster assignments, adjusted for chance. Given a contingency table between true labels and predicted clusters, ARI is computed as:

$$\text{ARI} = \frac{RI_{\text{obs}} - RI_{\text{rand}}}{\max(RI) - RI_{\text{rand}}} \tag{14}$$

where $RI_{\text{obs}}$ is the observed Rand index measuring the similarity between predicted and ground-truth clusterings, $RI_{\text{rand}}$ denotes the expected index under random labeling, and $\max(RI)$ is the maximum attainable value.

**Normalized Mutual Information (NMI)** [23]: Quantifies mutual dependence between true and predicted clusters. Defined as follows.

$$\text{NMI} = \frac{2 \cdot I(Y; \hat{Y})}{H(Y) + H(\hat{Y})}, \tag{15}$$

where $I(Y; \hat{Y})$ is the mutual information between true labels $Y$ and predicted labels $\hat{Y}$, and $H(\cdot)$ denotes entropy. Higher ARI and NMI values indicate better clustering alignment with biological ground truth.

## A.4 Extended Analysis for Main Performance (RQ1)

To supplement the main performance discussion in Section 4.2, we provide a detailed analysis of SAINT across all individual slices from the DLPFC, HBC, and MBA datasets, as reported in Table 4.

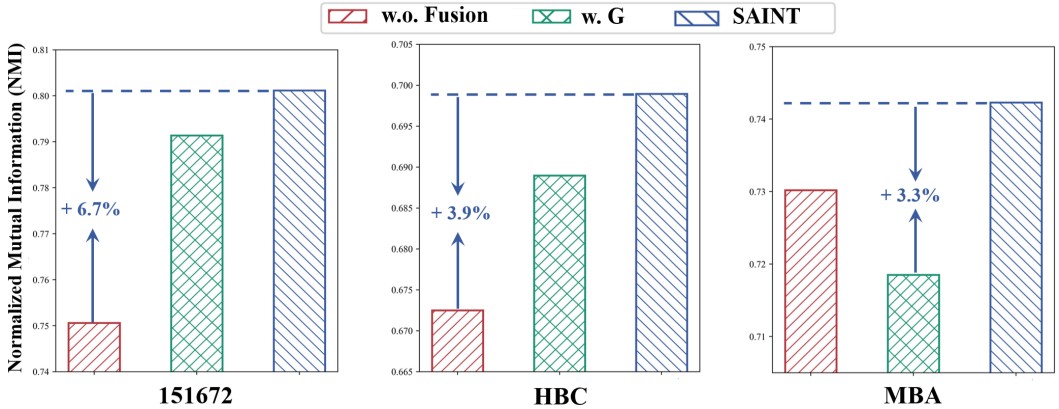

Figure 7: Ablation study of different components in SAINT.

Table 4: Full clustering performance of competing spatial transcriptomics models. Bold entries indicate the best results, and underlined values denote the second-best.

| Method | Adjusted Rand Index (ARI) | | | | | | | | | | | | | |
| | 151507 | 151508 | 151509 | 151510 | 151669 | 151670 | 151671 | 151672 | 151673 | 151674 | 151675 | 151676 | HBC | MBA |
|---|---|---|---|---|---|---|---|---|---|---|---|---|---|---|
| SCANPY | 0.20 | 0.15 | 0.19 | 0.14 | 0.10 | 0.09 | 0.12 | 0.12 | 0.20 | 0.22 | 0.23 | 0.22 | 0.49 | 0.23 |
| SpaGCN | 0.43 | 0.33 | 0.41 | 0.37 | 0.23 | 0.35 | 0.51 | 0.53 | 0.40 | 0.31 | 0.33 | 0.28 | 0.56 | 0.34 |
| DeepST | 0.55 | 0.42 | 0.43 | 0.50 | 0.44 | 0.33 | 0.52 | 0.48 | 0.54 | 0.55 | 0.53 | 0.56 | 0.53 | 0.25 |
| SCGDL | 0.49 | 0.34 | 0.32 | 0.31 | 0.24 | 0.26 | 0.31 | 0.34 | 0.33 | 0.27 | 0.30 | 0.29 | 0.35 | 0.26 |
| stLearn | 0.49 | 0.31 | 0.45 | 0.44 | 0.32 | 0.23 | 0.39 | 0.34 | 0.30 | 0.38 | 0.38 | 0.40 | 0.55 | 0.38 |
| Spatial-MGCN | 0.63 | 0.46 | 0.54 | 0.51 | 0.39 | 0.35 | 0.60 | 0.77 | 0.61 | **0.60** | 0.54 | 0.57 | 0.64 | 0.42 |
| GraphST | 0.48 | 0.49 | 0.52 | 0.50 | 0.48 | 0.46 | 0.61 | 0.63 | **0.63** | 0.43 | 0.55 | 0.55 | 0.54 | 0.41 |
| stMMR | 0.59 | 0.51 | 0.58 | 0.69 | 0.49 | 0.48 | 0.68 | 0.63 | 0.60 | 0.51 | 0.57 | 0.55 | 0.62 | 0.44 |
| MAFN | 0.68 | 0.51 | 0.71 | 0.61 | 0.56 | 0.48 | 0.82 | 0.76 | 0.57 | 0.50 | 0.46 | 0.53 | 0.60 | 0.43 |
| SAINT-G | 0.74 | 0.64 | 0.73 | 0.71 | 0.56 | 0.56 | 0.83 | 0.80 | 0.61 | 0.56 | 0.57 | 0.58 | 0.64 | 0.45 |
| SAINT-SA | **0.75** | **0.68** | **0.74** | **0.76** | **0.58** | **0.57** | **0.90** | **0.85** | 0.62 | 0.57 | **0.58** | **0.60** | **0.66** | **0.46** |
| Method | Normalized Mutual Information (NMI) | | | | | | | | | | | | | |
| | 151507 | 151508 | 151509 | 151510 | 151669 | 151670 | 151671 | 151672 | 151673 | 151674 | 151675 | 151676 | HBC | MBA |
| SCANPY | 0.21 | 0.21 | 0.27 | 0.22 | 0.16 | 0.16 | 0.24 | 0.23 | 0.29 | 0.31 | 0.32 | 0.31 | 0.52 | 0.45 |
| SpaGCN | 0.54 | 0.42 | 0.55 | 0.50 | 0.42 | 0.45 | 0.60 | 0.61 | 0.55 | 0.46 | 0.46 | 0.46 | 0.56 | 0.62 |
| DeepST | 0.62 | 0.57 | 0.62 | 0.62 | 0.57 | 0.51 | 0.59 | 0.60 | 0.69 | 0.69 | 0.66 | 0.68 | 0.68 | 0.57 |
| SCGDL | 0.55 | 0.44 | 0.48 | 0.45 | 0.38 | 0.36 | 0.41 | 0.46 | 0.42 | 0.38 | 0.41 | 0.42 | 0.43 | 0.64 |
| stLearn | 0.64 | 0.53 | 0.62 | 0.59 | 0.49 | 0.41 | 0.54 | 0.47 | 0.49 | 0.54 | 0.56 | 0.56 | 0.63 | 0.66 |
| Spatial-MGCN | 0.74 | 0.60 | 0.68 | 0.67 | 0.58 | 0.56 | 0.72 | 0.75 | 0.68 | **0.69** | 0.67 | 0.67 | 0.69 | 0.71 |
| GraphST | 0.64 | 0.54 | 0.64 | 0.64 | 0.59 | **0.68** | 0.70 | 0.61 | **0.74** | 0.61 | 0.62 | 0.66 | 0.67 | 0.71 |
| stMMR | 0.72 | 0.65 | 0.71 | 0.71 | 0.56 | 0.56 | 0.72 | 0.72 | 0.68 | 0.62 | 0.66 | 0.66 | 0.65 | 0.68 |
| MAFN | 0.74 | 0.51 | 0.72 | 0.68 | 0.63 | 0.60 | 0.78 | 0.75 | 0.67 | 0.62 | 0.60 | 0.67 | 0.67 | 0.73 |
| SAINT-G | 0.77 | 0.69 | 0.73 | 0.72 | 0.62 | 0.63 | 0.78 | 0.79 | 0.68 | 0.65 | 0.66 | 0.68 | 0.69 | 0.72 |
| SAINT-SA | **0.78** | **0.71** | **0.74** | **0.73** | **0.64** | 0.64 | **0.84** | **0.80** | 0.69 | 0.66 | **0.67** | **0.69** | **0.70** | **0.74** |

**Overall Trends.** SAINT-SA consistently achieves either the best or second-best performance across the vast majority of the 12 DLPFC slices. On average, it improves ARI from 0.64 (Spatial-MGCN) to 0.68 and NMI from 0.69 to 0.71. Notably, SAINT-Gdespite its simpler averaging-based sequence integrationalready outperforms all baselines in several slices, demonstrating the standalone benefits of incorporating gene-level nucleotide priors.

**DLPFC Dataset.** On DLPFC slices such as 151507, 151508, and 151509, SAINT-SA achieves ARI scores of 0.75, 0.68, and 0.74 respectivelyeach being the highest among all compared methods. Particularly, SAINT-G and SAINT-SA both outperform MAFN and GraphST by large margins. For instance, on slice 151671, SAINT-SA yields an NMI of 0.84, a +16.2% improvement over GraphST (0.72) and even surpasses MAFN by +7.7%. We also observe stable improvements on more challenging slices such as 151669 and 151670, where most baselines underperform (e.g., SCANPY < 0.1 ARI). SAINT-SA boosts ARI to 0.58 and 0.57, respectively, offering clear gains in low-signal scenarios.

**HBC Dataset.** The Human Breast Cancer (HBC) dataset features heterogeneous tumor microenvironments. Here, SAINT-G and SAINT-SA again dominate, reaching ARI scores of 0.64 and 0.66, and NMI scores of 0.70 and 0.70. These surpass MAFN by +3.1% ARI and +3.7% NMI. Compared to SpaGCN and DeepST, SAINT provides more spatially coherent and functionally aligned clustering, as further confirmed in qualitative case studies (see Appendix A.8).

**MBA Dataset.** On the more homogeneous mouse brain anterior (MBA) tissue, SAINT maintains strong performance, with ARI and NMI reaching 0.46 and 0.74. These results demonstrate the

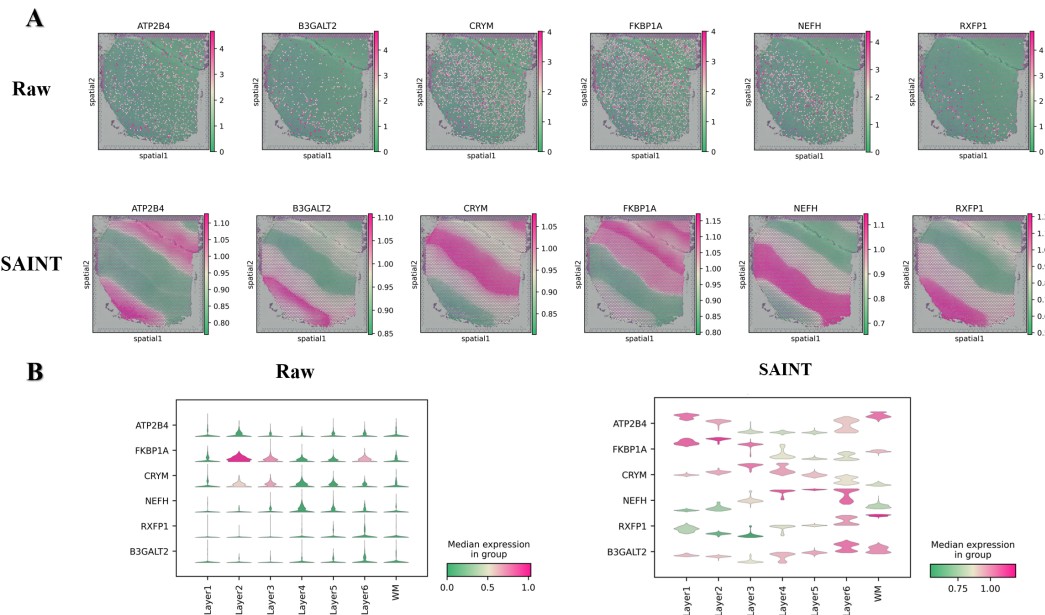

Figure 8: (A) Spatial expression of six marker genes before (Raw) and after SAINT enhancement. (B) Violin plots comparing layer-specific expression distributions of these genes in Raw and SAINT outputs.

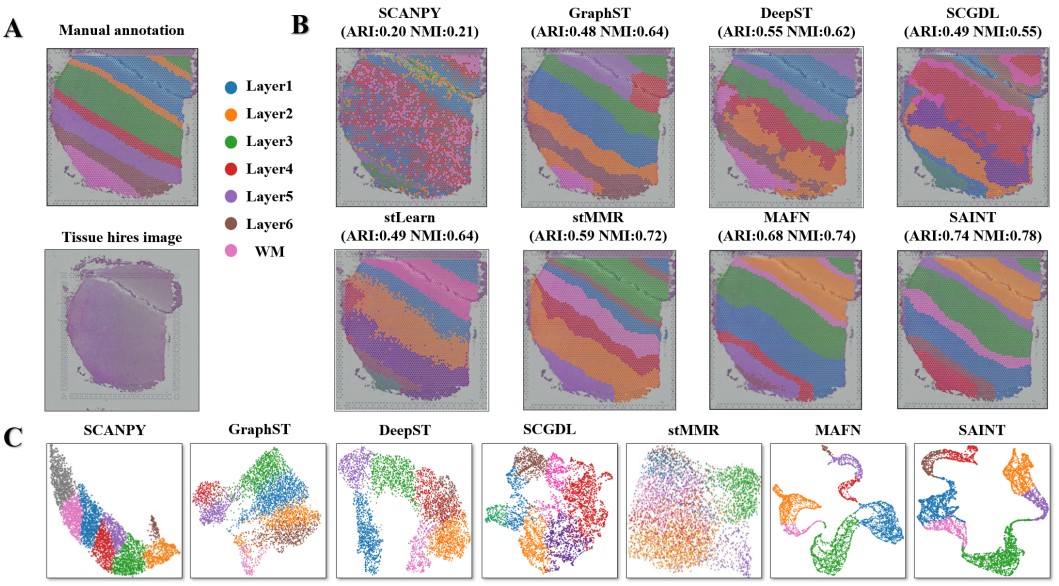

Figure 9: Case study of the proposed SAINT on DLPFC(151507) dataset.

generalizability of SAINT across diverse spatial contexts. The minimal drop in performance from HBC to MBA suggests that the framework is robust to differences in tissue type, gene expression scale, and biological variability.

Overall, these improvements reflect the effectiveness of nucleotide-informed representations in enhancing spatial transcriptomics clustering.

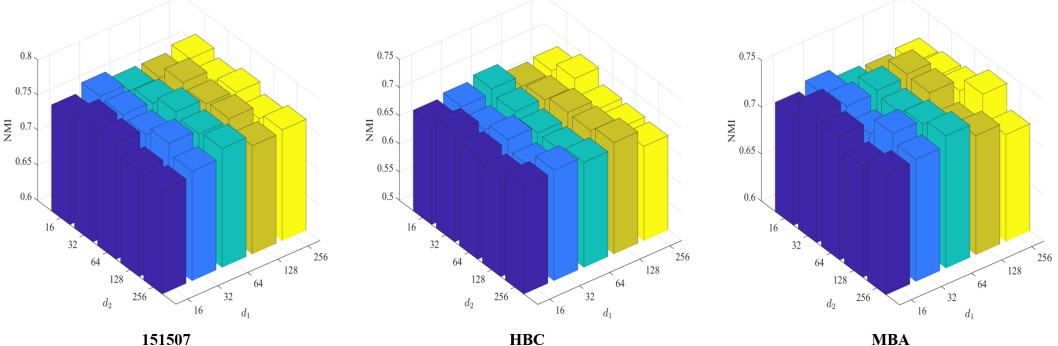

Figure 10: Sensitivity analysis of SAINT on NMI metrics.

## A.5 Detailed Ablation Study on NMI

To complement the ARI-based evaluation in the main text, we further analyze the NMI performance of different SAINT variants across three representative datasets: DLPFC 151672, HBC, and MBA. Figure 7 presents the corresponding NMI results.

Across all datasets, the full SAINT model achieves the highest NMI, validating the effectiveness of incorporating nucleotide-level priors through an attention-based fusion mechanism. On DLPFC slice 151672, SAINT improves over the baseline by +6.7% in NMI. Similarly, on HBC and MBA, relative gains of +3.9% and +3.3% are observed. Notably, even the average-pooling variant (denoted as w. SA) provides substantial improvements over the no-fusion baseline. For instance, on slice 151672, w. SA achieves a NMI of 0.79, compared to 0.75 for the baseline. These findings suggest that gene sequence embeddings capture biological context that complements spatial transcriptomic signals, even in the absence of attention. The consistent improvements across both tumor (HBC) and healthy (MBA, DLPFC) tissues further demonstrate that sequence priors enhance clustering generalizability in both homogeneous and heterogeneous spatial domains.

## A.6 Extended Sensitivity Analysis (NMI)

To complement the ARI-based sensitivity results, we further report the corresponding Normalized Mutual Information (NMI) scores under varying sequence embedding dimensions. As shown in Figure 10, we evaluate performance across a grid of values for $d_1$ and $d_2$ the embedding dimensions used in the SAINT-G and SAINT-SA modules, respectively chosen from $\{16, 32, 64, 128, 256\}$.

Across all three datasets (151507, HBC, and MBA), we observe that SAINT consistently maintains stable NMI scores under different dimension combinations. On slice 151507, NMI ranges from 0.7224 to 0.7386, with a relative variation of only 2.25%. For the HBC dataset, NMI varies from 0.6276 to 0.6550, corresponding to a 4.37% fluctuation. On the more structurally complex MBA dataset, scores range between 0.6212 and 0.6677, reflecting a 6.99% difference. These trends align with the ARI results and confirm that SAINT exhibits low sensitivity to the choice of sequence embedding dimensions. The model is capable of achieving strong performance across a broad range of $d_1$ and $d_2$ values without requiring delicate tuning.

## A.7 Impact of Sequence Embedding Dimension

To further investigate the effect of sequence embedding dimensionality on model performance, we conduct an ablation study by varying the dimensionality $d$ of nucleotide sequence embeddings across five values: $\{16, 32, 64, 128, 256\}$. The results, presented in Figure 6, report ARI and NMI scores on the DLPFC 151507 slice for three models: the original Spatial-MGCN, Spatial-MGCN+SAINT-G, and Spatial-MGCN+SAINT-SA.

We observe that both SAINT variants consistently outperform the base Spatial-MGCN across all dimensions. Notably, SAINT-SA achieves the best performance at $d = 32$ and $d = 64$, reaching an ARI of 73.57% and 72.38%, respectively. This corresponds to a relative ARI gain of +10.5% and +9.3% compared to SAINT-G at the same dimensions, and +16.7% and +15.0% over the baseline

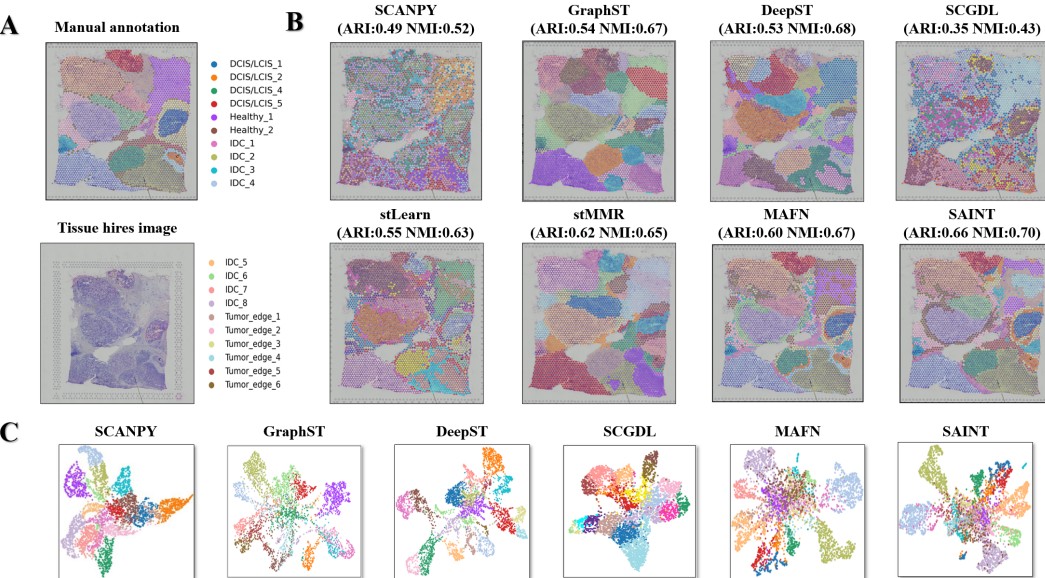

Figure 11: Case study of the proposed SAINT on HBC dataset.

Spatial-MGCN. Similarly, for NMI, SAINT-SA peaks at $d = 64$ with a score of 76.90%, surpassing SAINT-G and Spatial-MGCN by +0.96% and +2.47%, respectively. SAINT-G also consistently improves upon the base model, particularly at lower embedding sizes. At $d = 32$, SAINT-G achieves 71.25% ARI and 75.00% NMIgains of +8.2% (ARI) and +0.57% (NMI) over the base model. However, its performance tends to plateau or slightly decline at higher dimensions ($d = 128$ or $256$), suggesting potential overfitting or redundancy without the attention-guided fusion mechanism.

These findings indicate that the sequence-aware attention module in SAINT-SA enables more effective utilization of high-dimensional embeddings, maintaining robust and discriminative representations even as dimensionality increases. In contrast, the simple averaging strategy used in SAINT-G is more sensitive to dimensionality, showing diminishing returns beyond $d = 64$. Overall, the dimensional analysis validates the robustness and scalability of SAINT, especially in its full attention-based variant. It further confirms that integrating gene sequence representations can enhance spatial clustering performance in a dimension-aware manner.

## A.8  Case Study Analysis

To further investigate the biological interpretability of SAINT, we perform case studies on three representative spatial transcriptomics datasets (DLPFC, HBC, and MBA) to visually examine the clustering quality, gene expression continuity, and alignment with anatomical ground truth.

**DLPFC: Layer-specific Marker Recovery and Cortical Architecture**. Figure 8A shows the spatial expression patterns of six canonical marker genes*ATP2B4*, *B3GALT2*, *CRYM*, *FKBP1A*, *NEFH*, and *RXFP1*on slice 151507 before (Raw) and after (SAINT) model reconstruction. The raw data presents noisy, fragmented spatial signals. In contrast, SAINT smooths out spurious variations and reveals clear laminar boundaries that correspond well to cortical layers. In Figure 8B, violin plots further demonstrate improved expression stratification across layers. Genes like *ATP2B4* and *CRYM* exhibit sharper peaks in layer-specific distributions, indicating enhanced intra-cluster consistency. This suggests that SAINT captures biologically meaningful spatial organization that is obscured in raw measurements.

To further validate the spatial fidelity of SAINT, Figure 9 compares the clustering outputs of SAINT and six baseline methods on slice 151507 of the DLPFC dataset. Panel A displays the manually annotated ground truth and histological image for reference, while Panel B shows clustering results from competing models. As observed, classical methods such as SCANPY fail to capture spatial structure, resulting in noisy and biologically implausible regions. More advanced models like GraphST and DeepST partially recover cortical lamination but exhibit irregular boundaries or fragmented do-

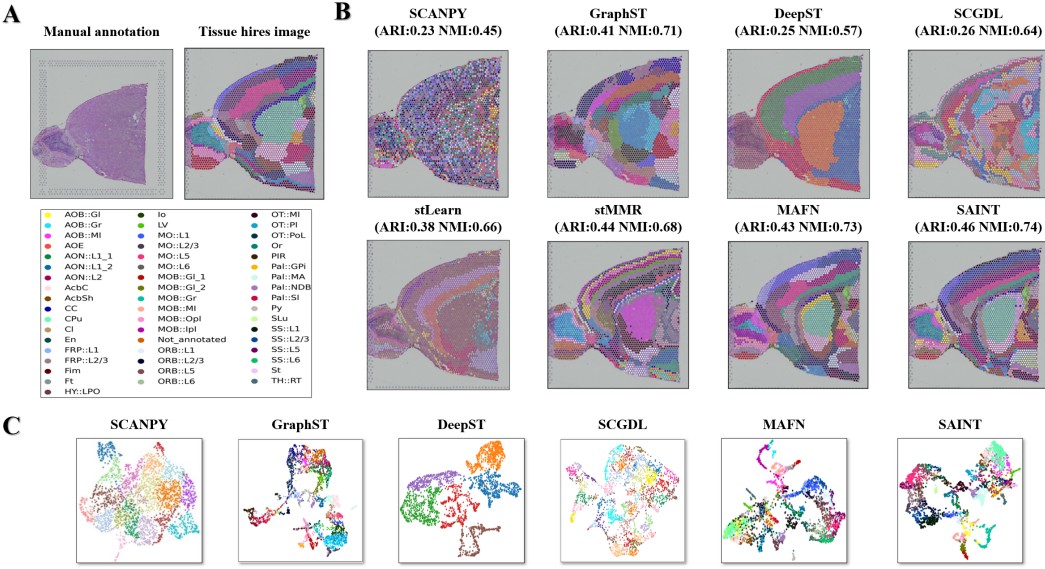

Figure 12: Case study of the proposed SAINT on MBA dataset.

mains. In contrast, SAINT produces well-aligned spatial clusters that closely match the manually defined cortical layers. Each layer is sharply delineated with minimal mixing, particularly in deeper layers such as Layer 5 and WM (white matter), where most other models show noticeable confusion. This suggests that the incorporation of nucleotide-level priors improves spatial coherence and biological interpretability. Panel C presents the 2D UMAP projections of spot embeddings. Compared to scattered or overlapping clusters generated by prior methods, SAINT yields more compact and clearly separable groups, further confirming its ability to preserve anatomical structure in the latent space.

**HBC: Tumor-edge Delineation and Microenvironment Disentanglement.**

As illustrated in Figure 11, SAINT shows improved resolution of complex tumor microenvironments in the HBC dataset. Compared to other methods that often over-fragment or blur tumor boundaries, SAINT delineates ductal carcinoma in situ (DCIS), invasive ductal carcinoma (IDC), and tumor-edge zones with higher coherence and spatial continuity. UMAP embeddings (Figure 11C) confirm this observation. Clusters identified by SAINT are compact, well-separated, and show clearer boundaries between IDC subtypes (e.g., $IDC_1$ to $IDC_8$). Notably, transitional tumor-edge areas are correctly positioned at cluster boundaries, reflecting subtle expression gradients across tissue regions. These results demonstrate SAINTs ability to model fine-grained spatial heterogeneity in complex pathological samples.

**MBA: Resolving Anatomical Hierarchies in the Mouse Brain.** In the MBA dataset (Figure 12), SAINT excels in reconstructing intricate anatomical subregions such as olfactory cortex (MOB), thalamus (TH), and hypothalamus (HY). While prior methods (e.g., SCANPY, SCGDL) tend to over-fragment or mix transitional areas, SAINT preserves spatial coherence and respects anatomical continuity. For instance, in regions like FRP::L2/3 and TH::RT, SAINT accurately recovers localized clusters that align with the brains hierarchical organization. The corresponding UMAP projection reveals compact and non-overlapping clusters, indicating that the learned embedding reflects both macro-structure and local transcriptional variation.

### A.9 Theoretical Complexity Analysis

The computational complexity of SAINT is mainly determined by the multi-view GCN layers, where each branch processes a feature matrix $X \in \mathbb{R}^{N \times F}$ with adjacency $A \in \mathbb{R}^{N \times N}$, resulting in $O(3|E|F)$, where $N$ is the number of spots, $F$ the feature dimension, and $|E|$ the number of edges. The sequence embedding branch only adds a lightweight projection from precomputed embeddings of dimension $d_s$ to $d_p$ with complexity $O(Nd_sd_p)$, and the ZINB decoder adds a negligible $O(Nd_p)$

term. Thus, the total complexity remains $O(|E|F)$, comparable to GCN-based methods like MAFN, with only a small linear overhead from the sequence branch.

## A.10    Comparison with Spatial-MGCN and MAFN

This study follows the experimental setup, datasets, and benchmarking protocols used in MAFN[62] to ensure fair comparison. The DLPFC, HBC, and MBA datasets, together with the same evaluation metrics, are adopted for consistency. Nevertheless, the proposed SAINT framework introduces several essential innovations beyond Spatial-MGCN[47] and MAFN[62] .

First, previous methods mainly integrate spatial and gene expression information through graph convolutional networks and adaptive fusion strategies, without considering the biological knowledge contained in gene nucleotide sequences. SAINT explicitly incorporates nucleotide-level representations into the clustering process, allowing biologically distinct genes with similar expression profiles to be differentiated.

Second, SAINT employs a sequence-aware encoder based on the pretrained Nucleotide Transformer, which converts gene sequences into high-dimensional embeddings that capture functional and regulatory genomic information not accessible from expression data alone.

Third, while Spatial-MGCN and MAFN both use expression and spatial graphs, SAINT integrates sequence-derived representations through a cross-modal decorrelation loss (DICR). This loss promotes complementary information across modalities rather than simple consistency.

Fourth, SAINT further contributes by constructing sequence-augmented datasets for widely used benchmarks, enabling reproducible evaluation of sequence-informed spatial models.

While the overall architectural design inherits certain effective components from previous framework, such as the graph embedding backbone and the zero-inflated negative binomial (ZINB) reconstruction loss, these elements serve as well-established foundations in spatial transcriptomics modeling. The ZINB loss is particularly suited to the sparse and overdispersed characteristics of gene expression data, offering better statistical fidelity than alternatives such as mean squared error (MSE). On top of this foundation, SAINT introduces biologically motivated enhancements, including attention-based aggregation that dynamically weights nucleotide features according to their relevance, thereby emphasizing informative signals and suppressing noise.

