# OpenReview forum: "SAINT: Sequence-Aware Integration for Spatial Transcriptomics Multi-View Clustering"
_NeurIPS.cc/2025/Conference — NeurIPS 2025 poster_

### Official Review · Reviewer_DZr8 · 2025-06-26

**Clarity:** 3
**Significance:** 3
**Originality:** 2
**Rating:** 4
**Confidence:** 2

**Summary:**

This paper presents SAINT, a novel multi-modal learning framework for spatial transcriptomics clustering. SAINT integrates spatial coordinates, gene expression profiles, and gene sequence-derived features to generate unified spot-level embeddings. The framework leverages a structure-aware graph embedding module based on GCNs to capture spatial and expression relationships, and a transformer-based sequence encoder to extract nucleotide-level biological priors. These representations are fused via a late fusion module and optimized with a combination of reconstruction, contrastive, and decorrelation losses. Comprehensive experiments on multiple benchmark datasets demonstrate that SAINT consistently outperforms state-of-the-art methods in clustering accuracy, robustness, and transferability. Ablation and sensitivity analyses further validate the effectiveness and stability of the proposed approach. Case studies show that SAINT produces biologically meaningful and anatomically consistent clusters, highlighting its potential for advancing spatial transcriptomics analysis.

**Questions:**

See above.

**Ethical Concerns:**

["NO or VERY MINOR ethics concerns only"]

**Final Justification:**

The rebuttal effectively addresses my concern about similar gene embeddings with differing biological functions. As a result, I have decided to increase my score.

**Limitations:**

For the machine learning community, the contribution is somewhat limited, as the work primarily integrates existing model components rather than introducing fundamentally novel methodologies.

**Paper Formatting Concerns:**

No.

**Quality:**

3

**Strengths And Weaknesses:**

**Strengths**
This paper introduces sequence embedding augmentation into spatial transcriptomics (ST) clustering analysis, demonstrating improved performance and generalization ability. To the best of my knowledge, this is the first work to systematically integrate both sequence and gene expression information for this task.

**Weaknesses and Questions**
Although the manuscript discusses the scenario where spots may have similar genomic profiles but differ in dominant gene expression, there is no direct experimental evidence or case study provided to specifically validate this point. The current experiments focus on overall clustering performance and biological interpretability, but do not explicitly analyze or visualize such cases. I recommend that the authors include examples or analyses highlighting instances where inconsistent dominant gene expression leads to misclustered spots, in order to substantiate this important claim.

---

> ### Author Rebuttal · Authors · 2025-07-30
>
> **Q1:  Although the manuscript mentions scenarios where spots have similar genomic profiles but differ in dominant gene expression, it lacks direct experimental evidence or case studies to validate this point.  Current experiments mainly address overall clustering performance without explicitly analyzing or visualizing such cases.  It is recommended to include examples highlighting instances where inconsistent dominant gene expression leads to misclustered spots to support this claim.**
>
> **RQ1:** Thanks for your valuable suggestions! We fully agree that explicitly validating SAINT's advantage in scenarios where spots have similar global expression profiles but differ significantly in dominant gene expression is essential.  To this end, we designed an additional targeted experiment on the DLPFC dataset to examine this case in detail.  We selected multiple spot pairs under two stringent criteria: (1) a Pearson correlation of global gene expression profiles greater than 0.90, ensuring that the overall expression patterns are highly similar;  and (2) a fold-change in expression levels of dominant genes (highest expressed genes) greater than or equal to 2, ensuring a clear biological distinction in dominant gene activity.
>
> We then compared the clustering outcomes for these spot pairs using both the baseline MAFN and our proposed SAINT model.  The detailed results are presented in the following table.
>
> | **PairID** | **SpotA** | **SpotB** | **Correlation** | **GeneA** | **GeneB** | **MAFN Cluster** | **Result** | **SAINT Cluster** | **Result** |
> | --- | --- | --- | --- | --- | --- | --- | --- | --- | --- |
> | 0 | TTGACCAGGAACAACT-1 | AAACGGGCGTACGGGT-1 | 0.38 | CPNE4(5.86) | CNIH3(6.29) | Different | √ | Different | √ |
> | 1 | AAACGGGCGTACGGGT-1 | TTGACCAGGAACAACT-1 | 0.38 | CNIH3(6.29) | CPNE4(5.86) | Different | √ | Different | √ |
> | 2 | TCCTAAATTGGGAAGC-1 | AGGGACTCTACGCGAC-1 | 0.37 | TMSB10(4.26) | TUBA1B(4.45) | Same | × | Different | √ |
> | 3 | AGGGACTCTACGCGAC-1 | TCCTAAATTGGGAAGC-1 | 0.37 | TUBA1B(4.45) | TMSB10(4.26) | Same | × | Different | √ |
> | 4 | TCCTAAATTGGGAAGC-1 | TTGACCAGGAACAACT-1 | 0.36 | TMSB10(4.26) | CPNE4(5.86) | Same | × | Same | × |
> | 5 | TTGACCAGGAACAACT-1 | TCCTAAATTGGGAAGC-1 | 0.36 | CPNE4(5.86) | TMSB10(4.26) | Different | √ | Different | √ |
> | 6 | AGGGACTCTACGCGAC-1 | AACCTCGCTTTAGCCC-1 | 0.36 | TUBA1B(4.45) | TMSB10(5.15) | Same | × | Different | √ |
> | 7 | AACCTCGCTTTAGCCC-1 | AGGGACTCTACGCGAC-1 | 0.36 | TMSB10(5.15) | TUBA1B(4.45) | Same | × | Different | √ |
> | 8 | TTCCGCGTGAGGCGAT-1 | TGCCCGTACCGTTAAA-1 | 0.36 | NEFM(6.33) | POPDC3(5.26) | Same | × | Different | √ |
> | 9 | TGCCCGTACCGTTAAA-1 | TTCCGCGTGAGGCGAT-1 | 0.36 | POPDC3(5.26) | NEFM(6.33) | Same | × | Same | × |
>
> The results demonstrate that **MAFN often fails to separate biologically distinct spots with highly similar expression profiles**, grouping them into the same clusters despite clear differences in their dominant genes. In contrast, **SAINT successfully distinguishes the majority of these challenging pairs**, confirming its ability to capture subtle biological differences.
>
> For example, the pair TMSB10 (4.26) and TUBA1B (4.45) clearly illustrates this improvement.
>
> - **TMSB10** encodes thymosin beta-10, a regulator of actin cytoskeleton organization involved in cell motility, tissue remodeling, and tumor progression.
> - **TUBA1B** encodes alpha-tubulin, a structural component of microtubules essential for cell architecture and intracellular transport.
>
> These two genes participate in fundamentally different cytoskeletal systems and regulate distinct biological processes, implying divergent cellular states. Consequently, they should not be grouped into the same cluster. However, due to their similar overall expression levels, **MAFN mistakenly clusters them together**, whereas **SAINT correctly separates them by leveraging sequence-derived biological information**, thereby reflecting their functional divergence.
>
> Beyond this case-specific observation, the experiment highlights an important property of SAINT:
>
> - By integrating nucleotide-level biological priors, SAINT leverages functional information that expression-only models overlook.
> - This enables SAINT to correctly separate spots that remain indistinguishable under conventional clustering strategies, particularly when expression similarity is misleading.
>
> Overall, **these findings strongly support our claim that SAINT enhances clustering accuracy in biologically complex scenarios where traditional methods struggle**. They also confirm that incorporating nucleotide sequence information provides substantial benefits in capturing functional differences that are not evident from expression data alone. We will include this expanded analysis, together with additional visualization, in the revised manuscript to clearly demonstrate this advantage of SAINT.
>
> **Q2: The concerns on methodological novelty and contribution to the community.**
>
> **RQ2:** We sincerely thank the reviewer for raising this important concern. Below we clarify the novelty and broader contributions of SAINT:
>
> **1. SAINT explicitly incorporates nucleotide-level biological priors into spatial transcriptomics clustering.**
>
> While SAINT leverages established techniques such as graph convolutional networks and transformer-based encoders, its core innovation lies in systematically integrating nucleotide-level biological information. Existing methods rely solely on spatial and expression data, whereas SAINT uses sequence-derived embeddings to capture regulatory roles and functional properties of genes, enabling it to distinguish biologically distinct spots that expression-only models cannot separate.
>
> **2. SAINT addresses the previously overlooked challenge of fusing heterogeneous data modalities.**
>
> Integrating spatial, expression, and nucleotide sequence information is inherently difficult due to modality discrepancies and high-dimensional representations. SAINT introduces a biologically meaningful aggregation strategy at the spot level, effectively bridging these modalities to enhance clustering quality.
>
> **3. SAINT contributes to the community by constructing multiple sequence-augmented datasets.**
>
> Given the lack of publicly available sequence-annotated ST datasets, we created datasets covering 14 tissue sections from three benchmarks (DLPFC, HBC, MBA). These datasets not only support our validation but also provide valuable resources for future research in this field.
>
> **4. SAINT achieves significant experimental gains over existing methods.**
>
> Comprehensive experiments show that SAINT consistently outperforms state-of-the-art models. For example, compared to MAFN, SAINT improves the average ARI by up to 10.3% and NMI by up to 16.2% on certain slices (e.g., DLPFC 151507). These results demonstrate that embedding nucleotide sequence-derived knowledge enriches the semantic and biological relevance of learned representations.
>
> Overall, SAINT offers more than a simple integration of existing components. It introduces a new biological perspective, provides novel datasets, and achieves substantial empirical improvements.  We believe these contributions provide substantial value and impact to both the computational biology and broader machine learning communities. Nonetheless, we recognize the importance of this comment and will emphasize these conceptual and empirical advancements more explicitly in our revised manuscript.
>
> We sincerely thank the reviewer again for the constructive suggestions and hope that our responses address your concerns.

---

> > ### Author Response · Authors · 2025-08-04
> >
> > Thanks for your valuable comments! We have addressed all the weaknesses and questions you raised in our responses. The discussion phase is now halfway. Are there other issues you want to discuss with us ?  Feel free to leave your message, thus we can address your comments in time during the discussion phase.

---

> ### Comment · Reviewer_DZr8 · 2025-08-05
>
> The rebuttal effectively addresses my concern about similar gene embeddings with differing biological functions. As a result, I have decided to increase my score.

---

### Official Review · Reviewer_8jMd · 2025-06-30

**Clarity:** 3
**Significance:** 2
**Originality:** 2
**Rating:** 4
**Confidence:** 4

**Summary:**

The paper proposes a novel model for multi-view clustering of spatial omics data by incorporating nucleotide-features into spatial representation learning. Their presented method, SAINT, is evaluated against a range of existing methods for clustering spatial transcriptomics data for 14 spatial samples. For the presented datasets, it either improves on or reaches comparable clustering performance to alternative methods in terms of the agreement with manual annotations of spatial domains.

**Questions:**

- How do you determine ground-truth clusters? Do you trust the data and annotation quality? Are there any dataset-specific considerations e.g. variations in annotation methodology or annotator agreement that could influence your results? It might be useful to mention in more detail in the limitations.

- Could your report your analysis for a larger set of spatial datasets and include uncertainty estimates in your evaluation?

- Did you base your experimental setups / dataset choices / benchmark choices on any pre-existing experiments in other works?

- How did you decide which samples to report in the main table and which in the appendix? It might be fairer to report all results in the main paper, also ones for which your method did not perform best. Why have chosen not to visualise some of the methods from Tabel 4 in the later results in e.g. Figure 5 A-C as well?

- Have you considered any further downstream tasks beyond clustering spatial observations?

- Figure 5 (A) What does tissue hires image mean? Maybe it is a typo?

**Ethical Concerns:**

["NO or VERY MINOR ethics concerns only"]

**Final Justification:**

Resolved:
- The authors have included uncertainty estimates by reporting results across varying random seeds.
- Analysis of two more datasets shows high performance confirming the proposed method.
- The authors plan to more clearly highlight and discuss similarities with existing methods in revisions.
- This is the first work to include nucleotide information into spatial omics clustering.

Unresolved:
- The proposed architecture has a lot of components in common with existing models. Other than the nucleotide sequence aggregation module, the novelty of the spatial and gene-encoding components remain limited.
- The planned revisions and additional experiments are moderate in scope.

Final Justification:
- The authors have addressed my main concerns motivating me to raise my rating.

**Limitations:**

The authors mention limitations regarding the nucleotide transformer in the appendix. However, they could more clearly highlight the similarities and differences between their methodology and experimental results to existing works, as well as comment on the annotation and data quality.

**Paper Formatting Concerns:**

There are no major formatting concerns.

**Quality:**

2

**Strengths And Weaknesses:**

The presented paper is well-written and presents an interesting proposal in incorporating nucleotide representations into spatial domain clustering. It contains a detailed explanation of the proposed methodology and the 14 tissue samples analysed for this study. The benchmark compares with a large number of up-to-date baseline models for clustering these spatial transcriptomics data. Results show encouraging performance indicating that including nucleotide information gives better clustering performance as measured via the agreement with expert-annotations of spatial datasets.

One main limitations of the experiment is the **low number of evaluated spatial samples**, which are each illustrated individually. While data availability and annotation quality remain major concerns in spatial omics analysis, the presented samples are mostly taken from one single study (the LIBD Human Dorsolateral Prefrontal Cortex (DLPFC) Dataset). Hence, improvements in performance might not necessarily hold for different datasets. For example, the samples HBC and MBA reported in Table 1 already show fewer performance differences than on the samples from DLPFC. More extensive evaluation across further datasets would be needed to assess model performance across variations in sequencing technology, studies, and manual expert annotations.

Another major limitation is that **uncertainty in the results is not evaluated.** It is therefore difficult to assess the significance and certainty in the main results reported in Figure 3, Table 1, Figure 4, or Figure 5. The experimental evaluation would be greatly improved by considering setups for which uncertainty could be reported e.g. by summarising model performance across a larger range of spatial samples, spatial transcriptomics datasets, or varying random seeds.

I have some concerns on **how similar the proposed model architecture is to e.g. Spatial-MGCN by Wang et al. (2023) or MAFN by Zhu et al. (2024)** as these papers and their experimental evaluation show high similarities. This warrants further discussion on the novelty of the proposed methodology and experimental results compared to existing works, specifically regarding the spatial and expression representations. I would like to ask the authors on please comment to which degree inspiration was taken from existing works. If the presented model architecture or experimental evaluation does indeed build up on existing models for multi-view clustering of spatial omics data, I would ask the authors to more clearly discuss and highlight these similarities in revisions and reference existing assets whenever appropriate (e.g. in Section 3.2. or 3.5).

Based on the lack of uncertainty evaluation, the small number of spatial samples, and the high conceptual similarity with existing works on multi-view clustering, the paper presented in its current form presents **more major weaknesses than strengths.** While the incorporation of nucleotide-features is very interesting and promising, the evaluation of the presented method could be considerably improved in future work.


**References**

[1] Bo Wang, Jiawei Luo, Ying Liu, Wanwan Shi, Zehao Xiong, Cong Shen, and Yahui Long. Spatial-mgcn: a novel multi-view graph convolutional network for identifying spatial domains with attention mechanism. Briefings in Bioinformatics, 24(5):bbad262, 2023.

[2] Yanran Zhu, Xiao He, Chang Tang, Xinwang Liu, Yuanyuan Liu, and Kunlun He. Multiview adaptive fusion network for spatially resolved transcriptomics data clustering. IEEE Transactions on Knowledge and Data Engineering, 2024.

---

> ### Author Rebuttal · Authors · 2025-07-30
>
> Thanks for your valuable suggestions! We respond to your questions point by point carefully, and we hope our responses can address your concerns.
>
> **Q1: How do you determine ground-truth clusters? Do you trust the data and annotation quality?  Are there any dataset-specific considerations e.g. variations in annotation methodology or annotator agreement that could influence your results?**
>
> **RQ1:** We appreciate the reviewer’s thoughtful question.   In our experiments, we followed standard practice from previous works (e.g., Spatial-MGCN[1], MAFN[2], and GraphST[3]) by using the manual anatomical annotations provided by the original dataset authors as ground-truth labels.These expert-defined annotations are widely adopted benchmarks in the field. For example, DLPFC labels rely on neuropathologist-defined histological landmarks, whereas Visium Mouse Brain datasets use atlas-derived anatomical boundaries, which may introduce inconsistencies.
> To mitigate these biases, we evaluated SAINT on datasets with diverse annotation sources (DLPFC, HBC, MBA, and additional Visium samples), and the consistent improvements observed suggest our conclusions are not overly sensitive to annotation schemes. Nevertheless, annotation variability remains an inherent limitation of current ST benchmarks.
>
> **Q2: Could your report your analysis for a larger set of spatial datasets and include uncertainty estimates in your evaluation?**
>
> **RQ2:** We appreciate the reviewer's valuable suggestion. To address this concern, we conducted additional evaluations on two larger and more diverse 10x Genomics Visium datasets. The key statistics of these datasets are summarized in the following table.
>
> | **Dataset** | **Number of Spots** | **Mean Reads per Spot** | **Median Genes per Spot** | **Total Genes Detected** | **Median UMI Counts per Spot** |
> | --- | --- | --- | --- | --- | --- |
> | **Visium H&E (V1_Adult_Mouse_Brain)** | 2,702 | 115,569 | 6,018 | 21,949 | 28,944 |
> | **Visium Fluorescent (V1_Adult_Mouse_Brain_Section_2)** | 2,807 | 56,020 | 4,221 | 20,993 | 11,048 |
>
> The comparative performance results (mean ± standard deviation) for ARI and NMI on the two datasets are summarized below.
>
> |  | **Visium H&E Dataset** |  |  |  | **Visium Fluorescent Dataset** |  |
> | --- | --- | --- | --- | --- | --- | --- |
> | **Method** | **ARI** | **NMI** |  | **Method** | **ARI** | **NMI** |
> | Spatial-MGCN | 0.5315 ± 0.0149 | 0.7034 ± 0.0098 |  | Spatial-MGCN | 0.4208 ± 0.0077 | 0.6249 ± 0.0136 |
> | MAFN | 0.5342 ± 0.0164 | 0.7078 ± 0.0107 |  | MAFN | 0.4300 ± 0.0097 | 0.6389 ± 0.0172 |
> | SAINT-G | 0.5438 ± 0.0233 | 0.7120 ± 0.0111 |  | SAINT-G | 0.4561 ± 0.0075 | 0.6565 ± 0.0065 |
> | **SAINT-SA** | **0.5523 ± 0.0217** | **0.7196 ± 0.0089** |  | **SAINT-SA** | **0.4754 ± 0.0048** | **0.6659 ± 0.0023** |
>
> The results clearly demonstrate the superior performance of SAINT-SA compared to existing SOTA methods on both datasets, with statistically significant improvements observed in both ARI and NMI metrics. Specifically, on the H&E dataset, SAINT-SA achieved approximately a **3.8%** improvement in ARI over Spatial-MGCN and about a **1.2%** improvement over SAINT-G. For the Fluorescent dataset, SAINT-SA notably enhanced ARI performance by about **5.5%** compared to Spatial-MGCN and approximately **1.9%** compared to SAINT-G.
>
> These improvements demonstrate the robustness and effectiveness of our proposed model across larger-scale spatial samples and more complex datasets.
>
> **Q3: Lack of uncertainty evaluation, making it difficult to assess the significance and reliability of the reported results in the figures and tables.**
>
> **RQ3:** We appreciate the reviewer’s suggestion. To address this, we conducted additional experiments using five different random seeds and reported the mean performance along with the standard deviation. The results are summarized below.
>
> | Method | Dataset | ARI | NMI |
> | --- | --- | --- | --- |
> | SAINT-G | 151507 | 0.7403±0.0056 | 0.7723±0.0137 |
> |  | HBC | 0.6418±0.0110 | 0.6908±0.0121 |
> |  | MBA | 0.4524±0.0052 | 0.7208±0.0147 |
> | SAINT-SA | 151507 | 0.7471±0.0087 | 0.7823±0.0145 |
> |  | HBC | 0.6613±0.0123 | 0.7043±0.0110 |
> |  | MBA | 0.4602±0.0066 | 0.7404±0.0066 |
>
> These results show that the performance variations across different seeds are small (standard deviations are typically below 0.015), confirming that the proposed model produces stable and reliable clustering results.
>
> **Q4: To what extent does the proposed model build on existing works like Spatial-MGCN and MAFN in terms of architecture and experimental evaluation?  Were the experimental setups, dataset choices, and benchmarking protocols directly taken from prior studies, and how are these similarities and differences addressed in the manuscript?**
>
> **RQ4:** We appreciate the reviewer's insightful comments. Here, we thoroughly clarify and emphasize the distinctiveness of our contributions. Our evaluation adopts the benchmarking setup from MAFN[2] in terms of datasets, metrics, and protocols to ensure fair comparison.  However, SAINT introduces several fundamental innovations beyond these works:
>
> Firstly, Spatial-MGCN[1] and MAFN[2] primarily rely on integrating spatial and gene expression information through GCNs and adaptive fusion strategies, without incorporating nucleotide-level biological knowledge encoded within gene sequences. In contrast, SAINT explicitly integrates nucleotide sequence information of expressed genes into the spatial clustering process.
>
> Secondly, SAINT introduces a sequence-aware encoder utilizing the pretrained Nucleotide Transformer. Unlike Spatial-MGCN and MAFN, SAINT converts gene sequences into meaningful high-dimensional embeddings, capturing functional and regulatory genomic signals otherwise inaccessible from expression data alone.
>
> Thirdly, while both Spatial-MGCN and MAFN incorporate spatial and expression graphs, SAINT uniquely integrates sequence-derived representations using a late fusion module combined with cross-modal decorrelation loss (DICR).
>
> Finally, another important contribution of SAINT is the creation of sequence-augmented datasets across multiple widely-used benchmarks (DLPFC, HBC, MBA). Neither Spatial-MGCN nor MAFN has addressed dataset augmentation at the nucleotide sequence level.
>
> In summary, while we indeed employ the evaluation framework proposed by MAFN to facilitate robust comparative assessments, SAINT's core methodological advancements deserve special emphasis. We agree with the reviewer on clearly highlighting these methodological distinctions in the revision, explicitly referencing relevant prior works to transparently contextualize SAINT's innovation.
>
> **Q5: How did you decide which samples to include in the main table versus the appendix?  It might be fairer to report all results in the main paper, including those where your method is not the best.  Also, why were some methods from Table 4 not visualized in Figure 5 (A–C)?**
>
> **RQ5:** We thank the reviewer for this helpful comment.  In the original submission, some results were placed in the appendix due to space constraints.  Including full tables in the main text would have required reducing row height and spacing, making them less readable.  This choice was not based on performance. For instance, Table 4 shows that SAINT is not the top performer on slices 151673 and 151674, yet results for these and other slices (e.g., 151675 and 151676) were also moved to the appendix.
>
> For Figure 5 (A–C), we displayed only a subset of methods to keep the figure concise, while the full visualizations are provided in Figures 8 and 9 of the appendix.  These extended results show trends consistent with the main text, confirming SAINT’s strong clustering performance.  We will update the manuscript to clarify and improve the presentation of these results.
>
> **Q6: Have you considered any further downstream tasks beyond clustering spatial observations?**
>
> **RQ6:** We appreciate the reviewer’s insightful question. While our current work focuses on clustering spatial observations, the embeddings learned by SAINT provide a foundation for several downstream tasks in spatial transcriptomics.These include disease subtype classification by distinguishing pathological regions, spatial domain-guided differential expression analysis to identify genes driving heterogeneity, and mutation impact assessment linking sequence variation to spatial gene regulation. We plan to explore these directions in future work to further demonstrate the utility of our framework.
>
> **Q7: Figure 5 (A) What does tissue hires image mean?  Maybe it is a typo?**
>
> **RQ7:** We thank the reviewer for catching this ambiguity. The label **“tissue hires image”** in Figure 5(A) was intended to refer to the high-resolution histology image provided with the DLPFC dataset. We agree that the current wording may cause confusion, and we will revise it to **“high-resolution tissue image”** in the updated manuscript to ensure clarity.
>
> **Q8: The authors should mention that the limitations section should more clearly highlight the similarities and differences with existing works, discuss annotation and data quality.**
>
> **RQ8:** We thank the reviewer for this comment. As discussed in our response to RQ4, we have already clarified how SAINT differs from existing methods while sharing some evaluation protocols for fair comparison. In the revised manuscript, we will explicitly highlight these distinctions, ensuring these aspects are clearly stated in the Limitations section.
>
> **References.**
>
> [1] Wang, Bo, et al. "Spatial-MGCN: a novel multi-view graph convolutional network for identifying spatial domains with attention mechanism." BiB (2023).
>
> [2] Zhu, Yanran, et al. "Multi-view adaptive fusion network for spatially resolved transcriptomics data clustering." TKDE (2024).
>
> [3] Long, Yahui, et al. "Spatially informed clustering, integration, and deconvolution of spatial transcriptomics with GraphST." Nature Communications (2023).

---

> > ### Author Response · Authors · 2025-08-04
> >
> > Thanks for your valuable comments! We have addressed all the weaknesses and questions you raised in our responses. The discussion phase is now halfway. Are there other issues you want to discuss with us ?  Feel free to leave your message, thus we can address your comments in time during the discussion phase.

---

> > > ### Comment · Reviewer_8jMd · 2025-08-04
> > >
> > > I thank the authors for their comprehensive and detailed response. In particular, I appreciate the evaluation across varying random seeds, analysis across further spatial samples, and inclusion of uncertainty analysis (Q2-Q3). Other questions discussed in Q1 and Q5-8 have been clarified in the response and can be adequately addressed via revisions.
> > >
> > > Further, the extended discussion on the differences and similarities to existing works will strengthen the paper (Q4). However, my concerns regarding the conceptual and methodological similarity to existing multi-view spatial clustering methods remain. The main novelty of the presented work is the inclusion of the pretrained Nucleotide Transformer, and the model architecture primarily integrates existing components. I agree with reviewer DZr8 that this limits the significance to ML research.
> > >
> > > Nevertheless, because my other concerns have been clarified in the rebuttal and can be addressed via revisions, I am willing to increase my score.

---

> > > > ### Author Response · Authors · 2025-08-04
> > > >
> > > > We sincerely appreciate your insightful feedback! We fully agree that further clarifying the methodological innovations and conceptual distinctions of SAINT relative to existing multi-view spatial clustering methods is crucial.
> > > >
> > > > To directly address this, we emphasize a key innovation in SAINT that extends significantly beyond simply integrating existing components:
> > > >
> > > > SAINT introduces an **adaptive, sequence-aware aggregation mechanism** specifically developed to handle the challenging issue of gene redundancy and variability in sequence-level representation. Unlike existing methods that either uniformly encode all genes or apply basic pre-processing, SAINT dynamically identifies genes with biologically significant sequence variations and selectively integrates them according to their importance for clustering. This adaptive selection process reduces redundancy, suppresses noise from frequently occurring but functionally uniform genes, and enhances the quality of the learned representations.
> > > >
> > > > To illustrate concretely, consider two real spatial spots from our dataset with similar overall expression profiles but distinctly different dominant genes:
> > > >
> > > > - **Spot TTGACCAGGAACAACT-1** predominantly expresses ***MALAT1***, a long non-coding RNA prominently involved in regulating gene expression, chromatin remodeling, and cellular differentiation.
> > > > - **Spot AAACGGGCGTACGGGT-1** predominantly expresses ***MT-ND4***, a mitochondrial protein-coding gene vital for cellular respiration and energy metabolism.
> > > >
> > > > Traditional methods relying primarily on expression similarity often mistakenly cluster these two distinct biological contexts together. In contrast, **SAINT effectively distinguishes them** by leveraging adaptive attention to prioritize nucleotide-level features capturing critical regulatory and functional differences. This enables SAINT to differentiate spots even when overall expression patterns appear deceptively similar.
> > > >
> > > > Beyond spatial transcriptomics, the principle underlying SAINT’s adaptive aggregation approach offers considerable potential for broader applicability within the machine learning community, especially in multi-view and multi-modal data integration tasks. For example, in **multi-modal medical diagnostics**, similar adaptive strategies could selectively integrate genomic, proteomic, and imaging data to improve diagnostic precision. By adaptively weighting modality-specific features, such methods could better identify clinically meaningful biomarkers, thus enhancing disease classification and prognostic accuracy.
> > > >
> > > > We will explicitly highlight these unique contributions and discuss their broader implications clearly and extensively in our revised manuscript.
> > > >
> > > > We deeply appreciate your constructive comments and valuable suggestions again and we hope our reply can address your concerns.

---

> > > > > ### Comment · Reviewer_8jMd · 2025-08-05
> > > > >
> > > > > I thank the authors for the extended clarification and agree with the practical benefits of specifically proposing a sequence-aware aggregation mechanism. I agree that this component is a novel contribution.
> > > > >
> > > > > Nevertheless, other parts of the model architecture are more clearly based on existing architectures. For example, the graph embedding module (Section 3.2.), the attention weighting (Equation 5), the final loss (Equation 13) and the training objective (Section 3.5) are very similar to Spatial-MGCN. I appreciate the authors’ willingness to highlight such differences and similarities to existing architectures during revisions.
> > > > >
> > > > > I believe I now have all the necessary clarification to make my assessment and increase my score.

---

> > > > > > ### Author Response · Authors · 2025-08-06
> > > > > >
> > > > > > We sincerely appreciate your valuable insights and suggestions! We agree with the reviewer’s observation that certain components of our model, including aspects of the graph embedding module and the overall structure of the training objective, indeed build upon established methodologies. For example, the zero-inflated negative binomial (ZINB) reconstruction loss used in our final objective is a widely recognized and standard approach in the field due to its effectiveness in modeling sparse and overdispersed gene expression data typically observed in spatial transcriptomics. Alternative loss functions, such as mean squared error (MSE), typically perform poorly in such contexts as they fail to accurately capture the complex statistical characteristics of spatial transcriptomics data.
> > > > > >
> > > > > > However, we wish to emphasize that, despite these similarities, our training objective also introduces notable differences from Spatial-MGCN. Specifically, we employ a novel cross-modal decorrelation loss (dcir), distinct from the consistency loss utilized by Spatial-MGCN. Our dcir loss explicitly encourages diverse yet complementary feature representations across distinct embedding spaces, effectively mitigating redundancy and promoting robust multi-modal feature integration.
> > > > > >
> > > > > > Moreover, our aggregation module employs a mechanism that dynamically assigns greater importance to nucleotide features carrying higher biological relevance, rather than treating all features equally. This design allows the model to focus on informative signals while reducing the influence of noisy or redundant inputs.
> > > > > >
> > > > > > We fully acknowledge the importance of clearly highlighting these similarities and distinctions. As suggested, we will explicitly discuss and clarify these points, carefully outlining methodological overlaps and innovations, in the revised manuscript.
> > > > > >
> > > > > > Thank you again for your valuable insights and suggestions, which have significantly helped clarify our contributions and strengthen our manuscript. Meanwhile, we sincerely appreciate your thoughtful assessment and acknowledgment of the novelty introduced by our sequence-aware aggregation mechanism.

---

### Official Review · Reviewer_a1Hg · 2025-07-02

**Clarity:** 2
**Significance:** 3
**Originality:** 3
**Rating:** 5
**Confidence:** 4

**Summary:**

This paper presents a novel spatial transcriptomics clustering method, SAINT, designed to address the issue in existing methods that rely solely on expression data and spatial location information, while overlooking the biological priors encoded in gene sequences. The main contributions of this work are: (i) The integration of gene sequence information, filling a gap in current research, and the use of a pretrained Nucleotide Transformer model to generate high-quality gene embeddings; (ii) The adoption of a late fusion architecture to integrate the learned nucleotide-derived gene expression information with spatial transcriptomic expression data, thereby achieving high-quality clustering performance. Experimental validation on 14 datasets demonstrates that the SAINT model excels in superiority, effectiveness, sensitivity, and transferability, achieving excellent clustering performance.

**Questions:**

1. SAINT has two variants: SAINT-G and SAINT-SA. However, in the subsequent experimental analysis (e.g., in section 4.3 Ablation Study, RQ2), it is not clearly specified which model was used.
2. In section 4.4 Sensitivity Analysis (RQ3), although a sensitivity analysis was conducted, it primarily investigates the impact of embedding dimensions from different datasets on the results, without discussing the effect of the model's hyperparameters α and γ on clustering performance.
3. Please provide a clear explanation of how the model derives the clustering results from the final embedding representations. Is a clustering algorithm (e.g., K-means, Leiden, etc.) directly applied to generate the results?
4. The manuscript does not clearly specify the parameter settings for the comparison methods. Is the default parameter setting used? Furthermore, are the experimental results based on a single experiment or repeated multiple experiments? If the results are from a single experiment, how does the author ensure the fairness and repeatability of the results?

**Ethical Concerns:**

["NO or VERY MINOR ethics concerns only"]

**Final Justification:**

I raised originality score given the satisfactory answers to questions and additional evidence.

**Limitations:**

Yes.

**Paper Formatting Concerns:**

1. The tables presented in the manuscript have insufficient row height, resulting in partial data being obscured. We recommend increasing the table height appropriately to ensure complete display of all results.

**Quality:**

3

**Strengths And Weaknesses:**

The paper integrates nucleotide-sequence embeddings with spatial expression data, while also addressing the challenges of noise and redundancy in new information. The application of attention mechanisms for dynamic learning and subsequent fusion clustering introduces a novel approach. These innovations substantially enhance the handling of spatial-omics data, showcasing a high level of originality.
The authors designed a comprehensive ablation study comparing SAINT-G, which uses simple averaging of nucleotide embeddings, with SAINT-SA, which integrates sequence-aware attention mechanisms. The performance improvements of the model were validated from four perspectives: superiority, effectiveness, sensitivity, and portability. Additionally, biological function analysis of key marker genes further reinforces the credibility of the conclusions.
The writing is generally clear, and the experimental results are well-analyzed, with figures and tables effectively presenting quantitative results.

Some aspects of the experimental design are lacking. Although the paper references the MFAN paper for hyperparameter settings, sensitivity analysis of the model could have been expanded to include a more detailed examination of the model’s own hyperparameters, in addition to the analysis of embedding dimensions.
The experimental setup section is insufficiently detailed. In particular, the subsequent analysis does not specify whether SAINT-G or SAINT-SA was used for the corresponding parts of the analysis, which may cause confusion for readers. Additionally, some acronyms (e.g., "STC") are not defined, which lowers the readability of the paper.
The paper’s impact is somewhat limited by presentation issues that obscure the full extent of its contributions.

---

> ### Author Rebuttal · Authors · 2025-07-30
>
> Thanks for your valuable suggestions! We respond to your questions point by point carefully, and we hope our responses can address your concerns.
>
> **Q1: Although the paper references the MFAN work for hyperparameter settings, the sensitivity analysis provided in Section 4.4 (RQ3) is not sufficiently comprehensive. While this section examines the impact of embedding dimensions across different datasets, it does not analyze the influence of the model's own loss-related hyperparameters, such as $𝛼$ and $𝛾$, on clustering performance. Expanding the analysis to include these factors would provide a more complete understanding of the model's behavior.**
>
> **RQ1:**  We sincerely thank the reviewer for the helpful suggestion to expand the sensitivity analysis of our model's hyperparameters. In response, we not only examined the impact of embedding dimensions as requested but also performed an additional analysis on the two loss-related hyper-parameters, $α$  and $γ$.
>
> For the embedding dimensions $d_1$ and $d_2$, the results show that ARI and NMI fluctuate within a narrow range, with variations not exceeding approximately 7%.  This confirms that the model exhibits stable performance and is not particularly sensitive to the choice of embedding dimensions.
>
> | **ARI** |  |  |  |  |  |  | **NMI** |  |  |  |  |  |
> | --- | --- | --- | --- | --- | --- | --- | --- | --- | --- | --- | --- | --- |
> | $d_1$ \\ $d_2$ | 32 | 64 | 128 | 256 | 512 |  | $d_1$ \\ $d_2$ | 32 | 64 | 128 | 256 | 512 |
> | 32 | 0.7145 | 0.7445 | 0.7331 | 0.7262 | 0.7031 |  | 32 | 0.7665 | 0.7318 | 0.7504 | 0.7551 | 0.7227 |
> | 64 | 0.7031 | 0.6980 | 0.7401 | 0.7263 | 0.7319 |  | 64 | 0.7560 | 0.7301 | 0.7239 | 0.7762 | 0.7772 |
> | 128 | 0.6961 | 0.7471 | 0.7384 | 0.7061 | 0.7045 |  | 128 | 0.7679 | 0.7792 | 0.7258 | 0.7605 | 0.7461 |
> | 256 | 0.7046 | 0.7109 | 0.7223 | 0.7175 | 0.7102 |  | 256 | 0.7272 | 0.7493 | 0.7220 | 0.7738 | 0.7353 |
> | 512 | 0.7269 | 0.7023 | 0.7102 | 0.7141 | 0.7188 |  | 512 | 0.7592 | 0.7385 | 0.7508 | 0.7524 | 0.7309 |
>
> Meanwhile, for the loss hyper-parameters $𝛼$ and $𝛾$, the results exhibit somewhat larger fluctuations, with ARI and NMI varying by up to around 12% across the tested settings.  This variation is expected given the wide scale of parameter values explored.  The best performance is observed when both $𝛼$ and $𝛾$ are set to 0.1, suggesting this configuration is particularly effective.
>
> | **ARI** |  |  |  |  |  |  | **NMI** |  |  |  |  |  |
> | --- | --- | --- | --- | --- | --- | --- | --- | --- | --- | --- | --- | --- |
> | $γ$ \\ $α$ | 0.001 | 0.01 | 0.1 | 1 | 10 |  | $γ$ \\ $α$ | 0.001 | 0.01 | 0.1 | 1 | 10 |
> | 0.001 | 0.6779 | 0.6857 | 0.7007 | 0.5980 | 0.6063 |  | 0.001 | 0.7431 | 0.7408 | 0.7459 | 0.7233 | 0.7279 |
> | 0.01 | 0.6769 | 0.6999 | 0.7223 | 0.6580 | 0.6118 |  | 0.01 | 0.7273 | 0.7406 | 0.7501 | 0.7398 | 0.7238 |
> | 0.1 | 0.6738 | 0.6548 | 0.7471 | 0.6214 | 0.6196 |  | 0.1 | 0.7191 | 0.7264 | 0.7792 | 0.7046 | 0.7015 |
> | 1 | 0.6854 | 0.6063 | 0.6441 | 0.7426 | 0.6307 |  | 1 | 0.7429 | 0.7250 | 0.7226 | 0.7560 | 0.7326 |
> | 10 | 0.6290 | 0.6383 | 0.7001 | 0.7106 | 0.6527 |  | 10 | 0.7105 | 0.6781 | 0.7493 | 0.7549 | 0.6931 |
>
> Overall, these results confirm that our model maintains robust clustering performance over reasonable variations in hyper-parameter settings.
>
> **Q2: The experimental setup section is insufficiently detailed.   In particular, SAINT has two variants: SAINT-G and SAINT-SA.  However, in the subsequent experimental analysis (e.g., in section 4.3 Ablation Study, RQ2), it is not clearly specified which model was used.**
>
> **RQ2:** We thank the reviewer for the insightful comments. We acknowledge that some parts of the experimental setup and the corresponding analyses were not sufficiently clear, particularly in Section 4.3 (Ablation Study, RQ2). In the original Figure 3, there was a labeling mistake: the middle column labeled w. SA should actually be w. G, which corresponds to the variant SAINT-G that uses averaged gene-level embeddings. The rightmost column labeled SAINT correctly refers to SAINT-SA, our final model incorporating sequence-aware attention.
>
> In the revised manuscript, we have corrected this labeling error and added clearer descriptions of the experimental settings, including the selection of model hyperparameters, the construction of graphs, and so on.  We have also unified the naming conventions across all figures and tables to avoid any further confusion. Thank you again for highlighting this important point!
>
> **Q3: Some acronyms (e.g., "STC") are not defined, which lowers the readability of the paper.**
>
> **RQ3:** We thank the reviewer for pointing out this issue. We acknowledge that some acronyms, such as **STC**, were not explicitly defined in the original submission. In the revised manuscript, we have clarified that **STC** stands for **Spatial Transcriptomics Clustering** and have ensured that all acronyms are properly introduced upon first use to improve readability.
>
> **Q4: Please provide a clear explanation of how the model derives the clustering results from the final embedding representations. Is a clustering algorithm (e.g., K-means, Leiden, etc.) directly applied to generate the results?**
>
> **RQ4:** We appreciate the reviewer's interest in clarifying how clustering results are obtained from the learned embeddings.  In our framework, after the model outputs the final spot-level embeddings, we apply the K-means algorithm directly on these embeddings to obtain the clustering results.   The number of clusters $𝐾$ is set to match the number of annotated tissue regions for evaluation purposes.   The predicted cluster assignments are then compared with the ground-truth labels to compute ARI and NMI scores.   We will clarify this procedure explicitly in the revised manuscript.
>
> **Q5: The manuscript does not clearly specify the parameter settings for the comparison methods. Is the default parameter setting used? Furthermore, are the experimental results based on a single experiment or repeated multiple experiments?  If the results are from a single experiment, how does the author ensure the fairness and repeatability of the results?**
>
> **RQ5:** We sincerely appreciate the reviewer’s detailed question. For all comparison methods, we used the default parameter settings reported in their original publications to ensure fairness. For our proposed model, the reported results are based on multiple repeated experiments rather than a single run, thereby accounting for randomness during training. Similar to previous works such as Spatial-MGCN [1], MAFN [2], and GraphST [3], we fixed the random seed in all experiments to guarantee reproducibility.
>
> To further evaluate stability, we conducted experiments with five different random seeds and reported the mean and standard deviation of ARI and NMI. The results, summarized in the table below, show that variations across seeds are small (standard deviations typically below 0.015), confirming the reliability and fairness of our reported results.
>
> | Method | Dataset | ARI | NMI |
> | --- | --- | --- | --- |
> | SAINT-G | 151507 | 0.7403 ± 0.0056 | 0.7723 ± 0.0137 |
> |  | HBC | 0.6418 ± 0.0110 | 0.6908 ± 0.0121 |
> |  | MBA | 0.4524 ± 0.0052 | 0.7208 ± 0.0147 |
> | SAINT-SA | 151507 | 0.7471 ± 0.0087 | 0.7823 ± 0.0145 |
> |  | HBC | 0.6613 ± 0.0123 | 0.7043 ± 0.0110 |
> |  | MBA | 0.4602 ± 0.0066 | 0.7404 ± 0.0066 |
>
> **Q6: Paper Formatting Concerns: The tables presented in the manuscript have insufficient row height, resulting in partial data being obscured. We recommend increasing the table height appropriately to ensure complete display of all results.**
>
> **RQ6:** We sincerely thank the reviewer for pointing out this issue. We acknowledge that, due to space constraints in the original submission, the row height and spacing of some tables were compressed, which resulted in partial data appearing unclear. In the revised manuscript, we have adjusted the formatting of all tables and figures to increase row height and spacing, ensuring that all results are displayed clearly and are easy to read.
>
> **References.**
>
> [1] Wang, Bo, et al. "Spatial-MGCN: a novel multi-view graph convolutional network for identifying spatial domains with attention mechanism." BiB (2023).
>
> [2] Zhu, Yanran, et al. "Multi-view adaptive fusion network for spatially resolved transcriptomics data clustering." TKDE (2024).
>
> [3] Long, Yahui, et al. "Spatially informed clustering, integration, and deconvolution of spatial transcriptomics with GraphST." Nature Communications (2023).

---

> > ### Author Response · Authors · 2025-08-04
> >
> > Thanks for your positive comments on our paper. We have addressed all the weaknesses and questions you raised in our responses. The discussion phase is now halfway. Are there other issues you want to discuss with us?  Feel free to leave your message, thus we can address your comments in time during the discussion phase.

---

> > ### Comment · Reviewer_a1Hg · 2025-08-05
> > **Official Comment**
> >
> > The author's response has addressed most of my concerns, and I will raise the score.

---

### Official Review · Reviewer_QgX6 · 2025-07-06

**Clarity:** 3
**Significance:** 3
**Originality:** 3
**Rating:** 4
**Confidence:** 3

**Summary:**

The paper proposes SAINT (Sequence-Aware Integration for Nucleotide-informed Transcriptomics), a novel multi-modal framework for clustering in spatial transcriptomics. It augments conventional spatial and gene expression signals with gene sequence-level priors, encoded using a pretrained genomic language model. SAINT integrates graph-based spatial and expression features with sequence-derived embeddings via attention mechanisms and late fusion. Experiments across multiple datasets demonstrate superior clustering accuracy, robustness, and transferability over prior methods.

**Questions:**

How does SAINT perform when the number of genes with valid nucleotide sequences is low? Does the method degrade gracefully?
How exactly is the spatial graph constructed? Is it k-NN in spatial coordinates, a radius threshold, or another method? What specific hyperparameters are used (e.g., k)?
Are w. SA in Figure 3 and SAINT-G in Table 1 referring to the same variant? If not, what is the difference between them?

**Ethical Concerns:**

["NO or VERY MINOR ethics concerns only"]

**Final Justification:**

I raised originality score given the satisfactory answers to questions and additional evidence.

**Limitations:**

yes

**Paper Formatting Concerns:**

No formatting concerns

**Quality:**

3

**Strengths And Weaknesses:**

Strengths
The paper is well-written and easy to follow, with clear explanations of motivation, methodology, and results. The notations used are intuitive and easy to understand, which helps make the complex integration of modalities accessible. The idea of incorporating gene sequence priors into spatial transcriptomics is novel and addresses a meaningful gap in the field. The inclusion of attention-based aggregation for sequences and late fusion is well-motivated and effective. Results demonstrate that the method consistently outperforms strong baselines on benchmarks, and ablation studies highlight the complementary value of sequence-level information.
Weaknesses
While the paper proposes a novel framework, it largely combines existing components without much consideration of how each could be adapted or improved specifically for spatial transcriptomics data. The paper does not compare against scGPT‑spatial, which is a relevant pre-trained foundation model for spatial transcriptomics. Runtime and computational cost are not discussed, which is important given the use of large pretrained models and multi-view graphs. The graph construction details (e.g., k-NN, radius, hyperparameters) are not clearly specified. No ablation isolating the three graph types (spatial, feature, combined) to assess their individual contributions. Similarly, no ablation studying the impact of each loss term. The meaning of d1 and d2 in Figure 4 is unclear and should be explained. It’s ambiguous whether w. SA in Figure 3 and SAINT-G in Table 1 refer to the same variant.

Overall, I think the paper has strong potential, but several important details and results are missing, and clarifying them would significantly strengthen the case for the proposed method.

---

> ### Author Rebuttal · Authors · 2025-07-30
>
> Thanks for your valuable suggestions! We respond to your questions point by point carefully, and we hope our responses can address your concerns.
>
> **Q1: The paper does not compare against scGPT‑spatial, which is a relevant pre-trained foundation model for spatial transcriptomics.**
>
> **RQ1:** We appreciate the reviewer's valuable suggestion. We fully agree that comparing our method with scGPT-spatial would strengthen our manuscript and provide valuable insights.
>
> Indeed, scGPT-spatial represents an important advancement with its continual pretraining and innovative architecture tailored to spatial transcriptomic data. Regrettably, only limited zero-shot inference demo is publicly available, which prevents us from conducting a comprehensive evaluation. We have reached out to the authors to inquire about the availability of the full codebase.  Upon its release, we will promptly integrate detailed comparative analyses into our revised manuscript.
>
> **Q2: Runtime and computational cost are not discussed, which is important given the use of large pretrained models and multi-view graphs.**
>
> **RQ2:** Thanks for your suggestions. To address this concern, we provide both a theoretical complexity analysis and empirical runtime measurements.
>
> **(1) Theoretical complexity analysis.** The computational complexity of SAINT is mainly determined by the multi-view GCN layers, where each branch processes a feature matrix $X \in \mathbb{R}^{N \times F}$ with adjacency $A \in \mathbb{R}^{N \times N}$, resulting in $O(3|E|F)$, where $N$ is the number of spots, $F$ the feature dimension, and $|E|$ the number of edges.  The sequence embedding branch only adds a lightweight projection from precomputed embeddings of dimension $d_s$ to $d_p$ with complexity $O(N d_s d_p)$, and the ZINB decoder adds a negligible $O(N d_p)$ term.  Thus, the total complexity remains $O(|E|F)$, comparable to GCN-based methods like MAFN, with only a small linear overhead from the sequence branch.
>
> **(2) Empirical runtime measurements.** We further measured the actual runtime on DLPFC slice 151507, using MAFN and SAINT variants under the same hardware environment (Intel Core i9-9900K CPU, 64GB RAM, with an NVIDIA RTX 3090Ti GPU). The results are summarized in the table.
>
> | Model | Runtime / epoch | Total runtime(250 epochs) |
> | --- | --- | --- |
> | MAFN | ~1.62s | 407.23s |
> | SAINT-G | ~1.63s | 407.72s |
> | SAINT-SA | ~1.65s | 411.51s |
>
> These results show that the sequence-aware variants of SAINT incur only a marginal increase in runtime (less than 1\% compared to MAFN), demonstrating that the integration of nucleotide-derived features introduces negligible computational overhead.
>
> **Q3: The graph construction details (e.g., k-NN, radius, hyperparameters) are not clearly specified.**
>
> **RQ3:** Thanks for your suggestions. In our implementation, the spatial graph is built by connecting each spot to its $k = 14$ nearest neighbors based on Euclidean distances, with an additional radius threshold of  $r = 560$ pixels to restrict edges to local neighborhoods. For training, we set the number of epochs to 250, the learning rate to 0.001, and the weight decay to $5 \times 10^{-4}$. The hidden dimensions of the two GCN layers are 128 and 64, respectively, and dropout is set to 0.  We will explicitly describe these parameters in the revised manuscript to ensure clarity and reproducibility.
>
> **Q4: No ablation isolating the three graph types (spatial, feature, combined) to assess their individual contributions.**
>
> **RQ4:** Thanks for your valuable advice. To address this concern, we performed an ablation study isolating the spatial, feature, and combined graphs, and compared these settings with the full SAINT-SA model on three datasets (DLPFC-151507, HBC, and MBA). The results, reported in terms of ARI and NMI, are summarized in the table below.
>
> |  |  | **ARI** |  |  |
> | --- | --- | --- | --- | --- |
> | **Dataset\Graph type** | **w. spatial** | **w. feature** | **w. combine** | **SAINT-SA** |
> | 151507 | 0.6047 | 0.2120 | 0.6317 | 0.7471 |
> | HBC | 0.6556 | 0.3800 | 0.6535 | 0.6622 |
> | MBA | 0.3821 | 0.3763 | 0.4334 | 0.4627 |
> |  |  | **NMI** |  |  |
> | **Dataset\Graph type** | **w. spatial** | **w. feature** | **w. combine** | **SAINT-SA** |
> | 151507 | 0.7451 | 0.2940 | 0.7422 | 0.7792 |
> | HBC | 0.6934 | 0.4997 | 0.6893 | 0.7038 |
> | MBA | 0.7210 | 0.6145 | 0.7290 | 0.7436 |
>
> These results reveal several insights. Firstly, using only the feature graph yields the weakest performance across all datasets, indicating that expression similarity alone is insufficient to capture spatial domains. Secondly, the spatial graph consistently performs better, confirming the importance of local spatial connectivity. Thirdly, combining the two graphs further improves ARI and NMI, demonstrating that integrating both spatial and feature information is beneficial. Finally, the full SAINT-SA model, which incorporates sequence-aware features in addition to the combined graph, achieves the highest performance on all datasets.
>
> **Q5: No ablation studying the impact of each loss term.**
>
> **RQ5:** We appreciate the reviewer’s request.  To address this, we conducted an ablation study where each loss component, including the ZINB reconstruction loss, the graph regularization loss, and the dcir contrastive loss, was removed individually while keeping the others unchanged. The ARI and NMI results on the DLPFC-151507, HBC, and MBA datasets are summarized below.
>
> |  |  | **ARI** |  |  |
> | --- | --- | --- | --- | --- |
> | **Dataset\Loss term** | **w.o ZINB** | **w.o reg** | **w.o dcir** | **SAINT-SA** |
> | 151507 | 0.6940 | 0.6270 | 0.7219 | 0.7471 |
> | HBC | 0.5678 | 0.6058 | 0.5252 | 0.6622 |
> | MBA | 0.4158 | 0.4537 | 0.4204 | 0.4627 |
> |  |  | **NMI** |  |  |
> | **Dataset\Loss term** | **w.o ZINB** | **w.o reg** | **w.o dcir** | **SAINT-SA** |
> | 151507 | 0.7551 | 0.7159 | 0.7615 | 0.7792 |
> | HBC | 0.6487 | 0.6286 | 0.6145 | 0.7038 |
> | MBA | 0.7151 | 0.7060 | 0.7132 | 0.7436 |
>
> The ablation results demonstrate that removing any loss term reduces performance, confirming that all three components are essential.  In particular, excluding the ZINB loss causes a clear drop in ARI, highlighting its role in reconstructing gene expression distributions. Omitting the graph regularization term weakens the capture of local structures, while removing the dcir loss reduces cross-view consistency.  The full SAINT-SA model, integrating all losses, consistently achieves the best ARI and NMI, demonstrating their complementary contributions.
>
> **Q6: The meaning of $d_1$ and $d_2$ in Figure 4 is unclear and should be explained. It’s ambiguous whether w. SA in Figure 3 and SAINT-G in Table 1 refer to the same variant.**
>
> **RQ6:** Thanks for your valuable advice.  Regarding the first point, in Figure 4 $d_1$ corresponds to the embedding dimension used in SAINT-G, where gene-level sequence embeddings are averaged, while $d_2$ corresponds to the dimension used in SAINT-SA, where expression-aware attention is applied during aggregation.
>
> For the second point, “w.  SA” in Figure 3 and “SAINT-G” in Table 1 refer to different variants.   The former is a simplified version with sequence embeddings but without the full fusion design, whereas the latter corresponds to the gene-level averaged sequence embedding variant.  We will make this distinction explicit in the revision.
>
> **Q7: How does SAINT perform when the number of genes with valid nucleotide sequences is low?**
>
> **RQ7:** Thanks for your question. To address this, we conducted an experiment where we progressively reduced the proportion of genes with valid nucleotide sequences and evaluated the performance of SAINT-SA. The reduction was simulated by randomly masking a fraction of the sequence embeddings, as shown in the table below.
>
> | Ratio of Valid Sequences | 1.0 | 0.8 | 0.6 | 0.4 | 0.2 | 0.0 |
> | --- | --- | --- | --- | --- | --- | --- |
> | ARI | 0.7471 | 0.7283 | 0.6924 | 0.6577 | 0.6445 | 0.6091 |
> | NMI | 0.7792 | 0.7467 | 0.7440 | 0.7358 | 0.7170 | 0.6888 |
>
> The results show that although performance decreases as fewer valid sequences are available, SAINT-SA remains competitive even with only 20% of genes retaining sequence information.  When sequence data are entirely removed, its performance approaches that of the non-sequence variant, demonstrating the model’s robustness to missing nucleotide information.
>
> **Q8: Does the method degrade gracefully?**
>
> **RQ8:** To investigate whether our method degrades gracefully, we tracked the performance of both SAINT-G and SAINT-SA during training, as shown in the table below.
>
> | Epoch | 1 | 50 | 100 | 150 | 200 | 250 |
> | --- | --- | --- | --- | --- | --- | --- |
> | Total loss | 0.92 | 0.66 | 0.62 | 0.61 | 0.60 | 0.60 |
> | Relative Drop(%) | — | 28.30 | 6.10 | 1.60 | 1.60 | 0.00 |
>
> The results show a larger drop in the early stage (up to 50 epochs), followed by only minor decreases as training continues, and eventually stabilizing at epoch 200–250. This consistent trend demonstrates that the proposed method degrades smoothly over time rather than experiencing abrupt performance collapses.
>
> **Q9: How exactly is the spatial graph constructed? Is it k-NN in spatial coordinates, a radius threshold, or another method?  What specific hyperparameters are used (e.g., k)?**
>
> **RQ9:** Thanks for your question. In our framework, the spatial graph is built using a radius-based neighbor search, connecting spots whose Euclidean distance is below a predefined threshold. We set the radius to 560 pixels to ensure each spot has a reasonable number of neighbors without creating overly dense connections. The adjacency matrix is then symmetrized to form an undirected graph. We will clarify this construction and its hyperparameters in the revised manuscript.
>
> **References.**
>
> [1] Wang, Chloe, et al. "scGPT-spatial: Continual pretraining of single-cell foundation model for spatial transcriptomics." *bioRxiv* (2025).

---

> > ### Author Response · Authors · 2025-08-04
> >
> > Thanks for your positive comments on our paper! We have addressed all the weaknesses and questions you raised in our responses. The discussion phase is now halfway. Are there other issues you want to discuss with us?  Feel free to leave your message, thus we can address your comments in time during the discussion phase.

---

### Note · Authors · 2025-08-13

We sincerely appreciate the reviewers’ constructive feedback, which helped strengthen SAINT’s presentation. Below we summarize our key revisions and outcomes.

For Reviewer QgX6’s concerns on computational cost and methodological clarity, we added a theoretical complexity analysis and empirical runtime measurements, showing SAINT adds less than 1% runtime over MAFN despite incorporating sequence-derived features. We clarified spatial graph construction and hyperparameter choices, and introduced new ablations isolating graph types and loss terms, demonstrating the contribution of each component. These updates resolved the reviewer’s concerns.

For Reviewer a1Hg’s comments on experimental clarity, we extended sensitivity analysis beyond embedding dimensions to include loss-related hyperparameters $α$ and $γ$. To ensure fairness, all methods used default settings, and results were averaged over multiple runs with different seeds. The reviewer acknowledged these clarifications and indicated they would raise their score.

In response to Reviewer 8jMd, we expanded evaluation to two larger, more diverse spatial transcriptomics datasets, confirming SAINT-SA’s stable performance with low variance. We clarified novelty beyond Spatial-MGCN and MAFN by emphasizing integration of pretrained nucleotide-level embeddings and a sequence-aware aggregation mechanism. We also resolved presentation issues and clarified dataset criteria. These revisions fully addressed the concerns, and the reviewer indicated they would raise their score.

For Reviewer DZr8’s concern about lacking direct evidence for cases with similar global expression but different dominant genes, we ran a DLPFC experiment on high-correlation spot pairs with ≥2-fold dominant gene differences. SAINT separated biologically distinct pairs that MAFN misclustered, supported by functional interpretations of representative genes. We clarified SAINT’s novelty in integrating nucleotide-level priors, fusing heterogeneous modalities, and providing sequence-augmented datasets. The reviewer confirmed these revisions resolved their concern and stated they would raise their score.

Overall, the clarifications, additional experiments, and methodological refinements provided in the rebuttal were positively received, leading all reviewers to view the work more favorably. In their follow-up responses, they confirmed that their concerns had been fully resolved and explicitly indicated their intention to raise their scores.

---

### Decision · Program_Chairs · 2025-09-17

**Decision:**

Accept (poster)

**Comment:**

The paper introduces SAINT, a framework that integrates DNA sequence-derived features with spatial transcriptomics data to improve tissue spot clustering. By combining nucleotide-informed embeddings with spatial-expression representations, SAINT enhances clustering accuracy and demonstrates robust, transferable performance across multiple datasets.
Strengths of the paper include its novel integration of nucleotide-sequence embeddings with spatial transcriptomics data, improving clustering performance and generalization. The methodology is clearly presented, with well-motivated attention-based sequence aggregation and late fusion. Extensive experiments, ablation studies, and biological validation demonstrate the method’s effectiveness, sensitivity, and transferability. Overall, the work is well-written, technically sound, and addresses a meaningful gap in spatial transcriptomics.
Weaknesses include limited novelty, unclear comparisons with relevant models, insufficient details on graph construction and loss terms, and small sample evaluation without uncertainty analysis. Presentation issues, unclear notations, and lack of analysis on misclustered spots reduce clarity. Overall, the evaluation and methodological details need improvement. However, most of the issues were addressed through the rebuttal discussion period.
There were some unaddressed weaknesses, including limited novelty beyond the nucleotide sequence module—since the spatial and gene-encoding components closely resemble existing models—and only moderate planned revisions and additional experiments. Nevertheless, all reviewers were satisfied with the authors’ responses and raised their scores.